# Transmembrane protein 135 regulates lipid homeostasis through its role in peroxisomal DHA metabolism

Michael Landowski [1,2], Vijesh J. Bhute [1,3], Samuel Grindel[1], Zachary Haugstad[1], Yeboah K. Gyening [4,5,6], Madison Tytanic [5,6], Richard S. Brush[5,6], Lucas J. Moyer[1], David W. Nelson[7], Christopher R. Davis[7], Chi-Liang Eric Yen[7], Sakae Ikeda[1,2], Martin-Paul Agbaga[4,5,6] & Akihiro Ikeda [1,2 ✉]

Transmembrane protein 135 (TMEM135) is thought to participate in the cellular response to increased intracellular lipids yet no defined molecular function for TMEM135 in lipid metabolism has been identified. In this study, we performed a lipid analysis of tissues from *Tmem135* mutant mice and found striking reductions of docosahexaenoic acid (DHA) across all *Tmem135* mutant tissues, indicating a role of TMEM135 in the production of DHA. Since all enzymes required for DHA synthesis remain intact in *Tmem135* mutant mice, we hypothesized that TMEM135 is involved in the export of DHA from peroxisomes. The *Tmem135* mutation likely leads to the retention of DHA in peroxisomes, causing DHA to be degraded within peroxisomes by their beta-oxidation machinery. This may lead to generation or alteration of ligands required for the activation of peroxisome proliferator-activated receptor a (PPARa) signaling, which in turn could result in increased peroxisomal number and beta-oxidation enzymes observed in *Tmem135* mutant mice. We confirmed this effect of PPARa signaling by detecting decreased peroxisomes and their proteins upon genetic ablation of *Ppara* in *Tmem135* mutant mice. Using *Tmem135* mutant mice, we also validated the protective effect of increased peroxisomes and peroxisomal beta-oxidation on the metabolic disease phenotypes of leptin mutant mice which has been observed in previous studies. Thus, we conclude that TMEM135 has a role in lipid homeostasis through its function in peroxisomes.

[1] Department of Medical Genetics, University of Wisconsin-Madison, Madison, WI, USA. [2] McPherson Eye Research Institute, University of Wisconsin-Madison, Madison, WI, USA. [3] Department of Chemical Engineering, Imperial College London, South Kensington, London, UK. [4] Department of Cell Biology, University of Oklahoma Health Sciences Center, Oklahoma City, OK, USA. [5] Department of Ophthalmology, University of Oklahoma Health Sciences Center, Oklahoma City, OK, USA. [6] Dean A. McGee Eye Institute, Oklahoma City, OK, USA. [7] Department of Nutritional Sciences, University of Wisconsin-Madison, Madison, WI, USA. ✉email: aikeda@wisc.edu

Dysregulation of lipid metabolism is a hallmark of many pathological conditions including diabetes[1], atherosclerosis[2], Alzheimer's disease[3], and age-related macular degeneration[4]. To maintain adequate regulation of lipid metabolism, cells depend on multiple organelles to balance the synthesis and breakdown of lipids. Notably, mitochondria serve as the regulatory hub for lipid metabolism[5]. Mitochondria possess the enzymatic machinery for beta-oxidation of fatty acids and conversion of these substrates to energy[6]. In conditions of excess energy production, mitochondria can transport excess metabolites to the cytosol for fatty acid synthesis that can curb fatty acid oxidation[7]. Furthermore, peroxisomes and endoplasmic reticulum (ER) aid mitochondria in their lipid metabolic duties. For example, peroxisomes exclusively degrade very long-chain and branched-chain fatty acids through beta- and alpha-oxidation[8] whereas the ER can degrade medium-chain fatty acids by omega-oxidation[9]. Peroxisomes can also generate 1-O-alkyl glycerol-3-phosphates for plasmalogen synthesis[10] and docosahexaenoic acid (DHA) from essential dietary fatty acids[8] while the ER produces membrane lipids[11]. Emerging evidence suggests that signaling occurs between organelles to preserve lipid homeostasis[12]. Thus, it is important to elucidate how organelles participate in lipid metabolism, which may lead to better treatments for diseases with disrupted lipid homeostasis.

Transmembrane protein 135 (TMEM135) is a 52 kilodalton protein with five transmembrane domains that is important for murine retinal and cardiac health[13–15]. Multiple proteomic studies have identified TMEM135 as a key component of peroxisomes[16–19], but it is also present in other organelles including mitochondria[13,20] and lipid droplets[20,21]. While no study has defined the molecular function of TMEM135 on these organelles, it has been speculated that TMEM135 may play a role in the cellular stress response to increased intracellular lipids[20]. This cellular stress response may impinge on mitochondrial dynamics[13,15,22–25], energy expenditure[20], and cholesterol degradation[26] as these pathways are affected in cells with altered TMEM135 function. In support of the hypothesis that TMEM135 participates in maintaining lipid homeostasis, we have recently found that a mutation in the murine Tmem135 gene increased the expression of genes involved in lipid metabolism in the murine retina[27], an organ with unique lipid demands[28].

In this study, we investigated the role of TMEM135 in lipid metabolism. We found that a mutation in the Tmem135 gene modified the lipid profiles of the livers, retinas, hearts, and plasmas of mice. Strikingly, we detected reductions in DHA in all tissues of the Tmem135 mutant (Tmem135^{FUN025/FUN025}) mice. Along with the decrease of DHA, we observed increased peroxisomal fatty acid oxidation enzymes and proliferation of peroxisomes that resulted from the activation of peroxisome proliferator-activated receptor a (PPARa) signaling in Tmem135 mutant mice. Furthermore, we confirmed the protective effect of the increases in peroxisomes and their beta-oxidation enzymes by observing amelioration of genetically-induced obesity, dyslipidemia, and fatty liver in mice due to the Tmem135 mutation. In summary, we conclude that TMEM135 is an important protein involved in lipid homeostasis through its function in peroxisomes.

## Results

### Tmem135 mutation reduces DHA concentrations in mice.
Previous studies have indicated TMEM135 has a role in lipid metabolism[20,26,27,29] but how TMEM135 contributes to maintaining lipid homeostasis remains unknown. In this study, we performed a high-throughput and semi-quantitative lipidomics analysis on tissues collected from mice with modified TMEM135 function. We evaluated the lipid profiles of four tissues (livers, hearts, retinas, and plasmas) from 2.5-month-old male wildtype (WT) mice, Tmem135^{FUN025/FUN025} mutant mice, and mice over-expressing Tmem135 (Tmem135 TG). Using principal components analysis, we found Tmem135^{FUN025/FUN025} tissues have distinct lipid profiles from WT and Tmem135 TG tissues (Fig. 1a). We focused on investigating the types of lipids altered in Tmem135^{FUN025/FUN025} tissues and observed many lipid classes were affected by the Tmem135 mutation (Supplementary Fig. 1, Supplementary Table 1). One common result in the Tmem135^{FUN025/FUN025} tissues was decreased membrane lipids classified as phosphatidylcholine (PC) and phosphatidylethanolamine (PE) (Supplementary Fig. 1, Supplementary Table 1). Remarkably, the Tmem135^{FUN025/FUN025} livers had accumulations of triglycerides (Supplementary Fig. 1a) while the Tmem135^{FUN025/FUN025} plasmas had reductions of triglycerides (Supplementary Fig. 1b) that appeared to be unique features of these Tmem135^{FUN025/FUN025} tissues. Next, we studied the acyl side chains of the lipids that were significantly different in Tmem135^{FUN025/FUN025} tissues (Supplementary Table 2). We found a large proportion of lipids containing DHA (C22:6n3) decreased across all the Tmem135^{FUN025/FUN025} tissues used in this study (Fig. 1b). We also observed modifications of lipids containing other fatty acids including C16:0, C16:1, C18:0, C18:1, C18:2, C20:3, C20:4, and C22:5 in the Tmem135^{FUN025/FUN025} tissues (Supplementary Table 2). However, the changes in these lipids appeared to occur in a tissue-specific manner (Supplementary Table 2). In summary, our lipidomics analysis revealed significant alterations in tissue lipid profiles and reductions of DHA-containing lipids as a common consequence of the Tmem135 mutation in mice.

To confirm the decreases of DHA-containing lipids in Tmem135^{FUN025/FUN025} tissues, we quantified the concentrations of fatty acids in the livers, retinas, hearts, and plasmas of 2.5-month-old WT and Tmem135^{FUN025/FUN025} mice. We detected significant alterations in the fatty acid composition of the different tissues from the Tmem135^{FUN025/FUN025} mice (Supplementary Table 3). We found that DHA was robustly decreased in Tmem135^{FUN025/FUN025} livers (Fig. 2a), retinas (Fig. 2b), hearts (Fig. 2c), and plasmas (Fig. 2d) relative to WT controls. Thus, results from our fatty acid analysis confirmed that the Tmem135 mutation reduces DHA concentrations in multiple murine tissues.

### Reduced DHA in Tmem135 mutant mice is not due to defects in the Sprecher pathway of DHA synthesis.
Because reductions in DHA were a common observation in the Tmem135^{FUN025/FUN025} tissues, we focused on investigating the role of TMEM135 in DHA metabolism. In vivo, DHA is derived from either the conversion of alpha-linolenic acid to DHA within cells via the "Sprecher pathway" or through the consumption of foods enriched in DHA[30]. Since the diets fed to the mice used in this study do not contain DHA, all DHA present in these mice must originate from endogenous DHA biosynthesis. Although all tissues have the capacity to generate DHA, the main producer of this PUFA species within the mouse is the liver[31]. To decipher the role of TMEM135 in cellular DHA metabolism, we harvested livers from 2.5-month-old WT and Tmem135^{FUN025/FUN025} mice and determined the expression level of key components of the Sprecher pathway of DHA synthesis (Fig. 3a)[32]. We performed qPCR for the desaturases [fatty acid desaturase 1 (Fads1) and 2 (Fads2)] and elongases [elongation of very-long-chain fatty acids-like 2 (Elovl2) and 5 (Elovl5)] required for the desaturation and elongation of dietary essential fatty acid 18:3n3 to generate C24:6n3 in the ER[33]. We found no significant changes in the expression

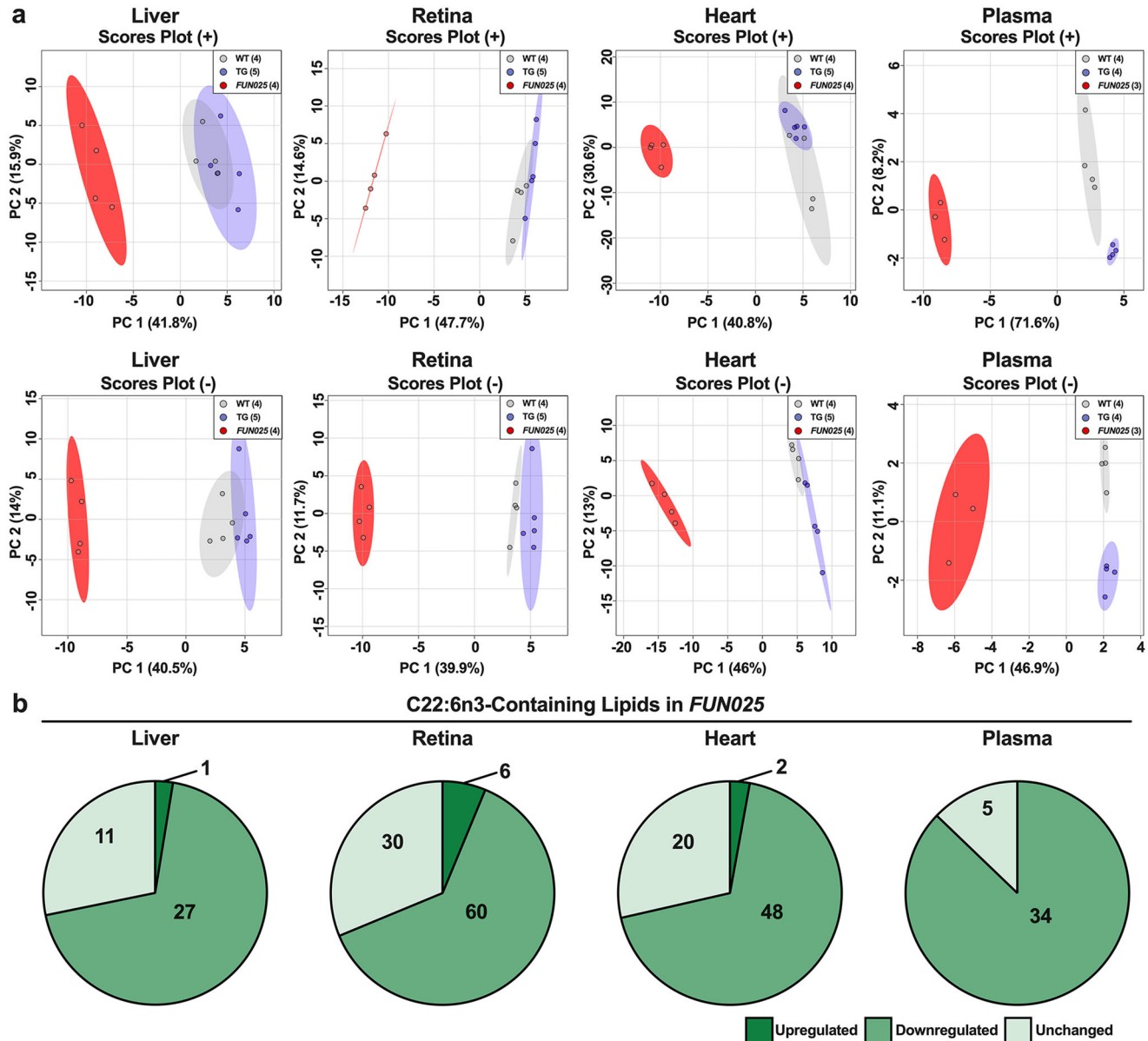

**Fig. 1 Docosahexaenoic polyunsaturated fatty acid-containing lipids are reduced in *Tmem135* mutant tissues. a** Principal component analysis of lipids detected in the positive ($+$) ion mode and negative ($-$) ion mode of male WT, *Tmem135* TG (TG), and *Tmem135*$^{FUN025/FUN025}$ (*FUN025*) livers, retinas, hearts, and plasmas based on log10 lipid concentrations. Number in parentheses represent the $N$ of independent mouse samples per genotype used in the experiment. **b** Pie graphs of docosahexaenoic acid (DHA, C22:6n3)-containing lipids that were significantly upregulated (dark green) or downregulated (medium green) as well as unchanged (light green) in *Tmem135*$^{FUN025/FUN025}$ tissues compared to WT. Numbers denote the total of lipids within the category. All altered lipid species can be found in Supplementary Tables 6–9. Significance was determined by one-way ANOVA with post hoc Tukey's test between WT and *Tmem135*$^{FUN025/FUN025}$ livers ($P < 0.05$).

levels of *Fads1*, *Fads2*, *Elovl2*, and *Elovl5* in *Tmem135*$^{FUN025/FUN025}$ livers relative to WT (Fig. 3b).

We also performed qPCR for ATP binding cassette subfamily D member 2 (*Abcd2*) which encodes for the importer of C24:6n3 into the peroxisome[34]. However, we detected no significant difference in *Abcd2* expression in *Tmem135*$^{FUN025/FUN025}$ livers relative to WT (Fig. 3c). Next, we utilized Western blot analysis to quantitate peroxisomal beta-oxidation enzymes required for generating C22:6n3 from C24:6n3: acyl-Coenzyme A oxidase 1 (ACOX1), D-bifunctional protein (DBP), acetyl-Coenzyme A acyltransferase 1 (ACAA1), and sterol carrier protein x (SCPx)[35]. We found a significant increase in ACOX1, DBP, and ACAA1 but no significant change in SCPx between *Tmem135*$^{FUN025/FUN025}$ and WT livers (Fig. 3d). Our data shows that the peroxisomal

beta-oxidation enzymes of the Sprecher pathway of DHA synthesis are increased in the *Tmem135*$^{FUN025/FUN025}$ livers. Taken together, our data suggest that reduced DHA in the *Tmem135* mutant mice does not result from a defect in previously-identified components of the Sprecher pathway of DHA synthesis.

**Tmem135 mutation increases peroxisomes in mice.** Next, we investigated if the upregulation of peroxisomal beta-oxidation enzymes in *Tmem135*$^{FUN025/FUN025}$ livers was due to an increase in peroxisomes. We labeled WT, *Tmem135*$^{FUN025/FUN025}$, and *Tmem135* TG liver cryosections with an anti-peroxisome biogenesis factor 14 (PEX14) antibody to visualize hepatic peroxisomes (Fig. 4a). After quantifying peroxisomes using a previously

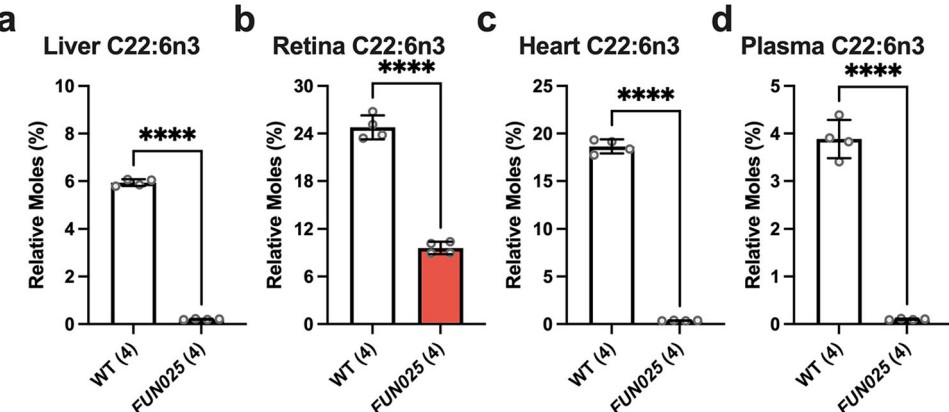

**Fig. 2 Docosahexaenoic polyunsaturated fatty acid concentrations are reduced in *Tmem135* mutant tissues.** Bar graphs depicting relative moles of docosahexaenoic acid (DHA, C22:6n3) quantified by gas chromatography mass spectrometry (GC-MS) in four 2.5-month-old wildtype (WT) (3 male/1 female) and *Tmem135^{FUN025FUN025}* (FUN025) (2 male/2 female) livers (**a**), retinas (**b**), hearts (**c**), and plasmas (**d**). All fatty acid data can be found in Table S3. **** indicates $P < 0.0001$ significance by two-way Student's $t$ test. Number in parentheses represents the $N$ of independent mouse samples per genotype used in the experiment. Dots represent individual data points. Data are presented as mean ± SD.

described method[36], we found an increase of peroxisomes in the *Tmem135^{FUN025/FUN025}* livers while there was a decrease of peroxisomes in the *Tmem135* TG livers (Fig. 4b). We validated our immunohistochemistry results through Western blot analysis probing for PEX14 (Fig. 4c). We also confirmed these findings by examining the expression of two additional peroxisome markers, peroxisome membrane protein 70 (PMP70) (Supplementary Fig. 2) and catalase (CAT) (Supplementary Fig. 3), in the livers of WT, *Tmem135^{FUN025/FUN025}*, and *Tmem135* TG mice. Next, we investigated if TMEM135 regulates peroxisomal proteins in other tissues of *Tmem135^{FUN025/FUN025}* and *Tmem135* TG mice. We examined the expression level of PMP70 in the neural retinas, eyecups, and hearts of *Tmem135^{FUN025/FUN025}* and *Tmem135* TG mice since these tissues displayed decreases of DHA in *Tmem135^{FUN025/FUN025}* mice. We detected a significant increase of PMP70 in the *Tmem135^{FUN025/FUN025}* neural retinas, eyecups, and hearts while there were decreases of PMP70 in the *Tmem135* TG neural retinas, eyecups, and hearts (Supplementary Fig. 4). We further confirmed that the peroxisome number is increased by the *Tmem135* mutation using cultured fibroblasts from WT, *Tmem135^{FUN025/FUN025}*, and *Tmem135* TG mice. We performed immunocytochemistry on these cells using an antibody targeting PEX14 to count the number of peroxisomes per cell (Fig. 5a). We found more peroxisomes in *Tmem135^{FUN025/FUN025}* fibroblasts and fewer peroxisomes in *Tmem135* TG fibroblasts compared to WT fibroblasts (Fig. 5b). We also detected similar changes in PMP70 protein levels between cultured fibroblasts from these genotypes (Supplementary Fig. 5). Our results show an important role of TMEM135 function in maintaining the number of peroxisomes in multiple mouse tissues and cell types.

**Tmem135 mutation increases PPARa signaling in the murine liver.** The increase in peroxisomes observed in the *Tmem135^{FUN025/FUN025}* tissues and cells could result from the activation of a peroxisome proliferator-activated receptor (PPAR)[37]. To determine the contribution of PPAR signaling on the peroxisomal changes in the *Tmem135^{FUN025/FUN025}* mice, we crossed *Tmem135^{FUN025/FUN025}* mice with peroxisome proliferator-activated receptor alpha knockout (*Ppara^{-/-}*) mice to generate *Tmem135^{FUN025/FUN025}/Ppara^{-/-}* mice that do not express one of the main PPAR proteins in the liver[38]. We performed immunohistochemistry on liver sections from WT, *Ppara^{-/-}*, *Tmem135^{FUN025/FUN025}*, and *Tmem135^{FUN025/FUN025}/Ppara^{-/-}*

mice for quantification of the number of PEX14-positive peroxisomes in order to determine the effect of PPARa on peroxisome proliferation due to the *Tmem135* mutation (Fig. 6a). Our quantification indicated reduced peroxisomes in *Tmem135^{FUN025/FUN025}/Ppara^{-/-}* livers compared to *Tmem135^{FUN025/FUN025}* livers (Fig. 6b), indicating that activation of PPARa is responsible for peroxisome proliferation in *Tmem135^{FUN025/FUN025}* mice. We also examined the expression of additional peroxisomal proteins (PMP70, ACOX1, DBP, and ACAA1) using liver lysates from WT, *Ppara^{-/-}*, *Tmem135^{FUN025/FUN025}*, and *Tmem135^{FUN025/FUN025}/Ppara^{-/-}* mice by Western blot (Fig. 6c). We found that the concentrations of PMP70, ACOX1, DBP, and ACAA1 were lower in *Tmem135^{FUN025/FUN025}/Ppara^{-/-}* livers than in *Tmem135^{FUN025/FUN025}* livers (Fig. 6c). Interestingly, the levels of PMP70 and ACOX1 in *Tmem135^{FUN025/FUN025}/Ppara^{-/-}* livers remained significantly higher than WT livers (Fig. 6c), suggesting PPARa-independent pathways may also contribute to the increases of PMP70 and ACOX1 in the *Tmem135^{FUN025/FUN025}* livers. To further confirm the activation of PPARa signaling in the *Tmem135^{FUN025/FUN025}* livers, we measured the level of cytochrome P450, family 4, subfamily a, polypeptide 10 (CYP4A10), a protein whose expression is contingent on PPARa[39], in these same liver lysates (Supplementary Fig. 6). We detected a significant increase of CYP4A10 in *Tmem135^{FUN025/FUN025}* livers compared to WT controls, while the CYP4A10 level in *Tmem135^{FUN025/FUN025}/Ppara^{-/-}* livers is comparable to WT controls (Supplementary Fig. 6), supporting the activation of PPARa signaling in the mouse liver due to the *Tmem135* mutation. In summary, our data shows the increase of peroxisomes and their proteins in the *Tmem135* mutant liver is at least partly mediated by PPARa signaling.

**Tmem135 mutation reduces obesity and dyslipidemia phenotypes in leptin mutant mice.** Having established an important role of TMEM135 in peroxisomal homeostasis, we sought to determine the physiological significance of peroxisomal changes due to the *Tmem135* mutation in a mouse model with dysregulated lipid metabolism. We chose leptin mutant (*Lep^{ob/ob}*) mice because modifications in peroxisomal function are known to modulate the phenotypes of *Lep^{ob/ob}* mice[40,41]. In order to test whether the peroxisomal changes caused by the *Tmem135* mutation have an effect on any of the disease phenotypes in the *Lep^{ob/ob}* mice, we crossed *Tmem135^{FUN025/FUN025}* mice with

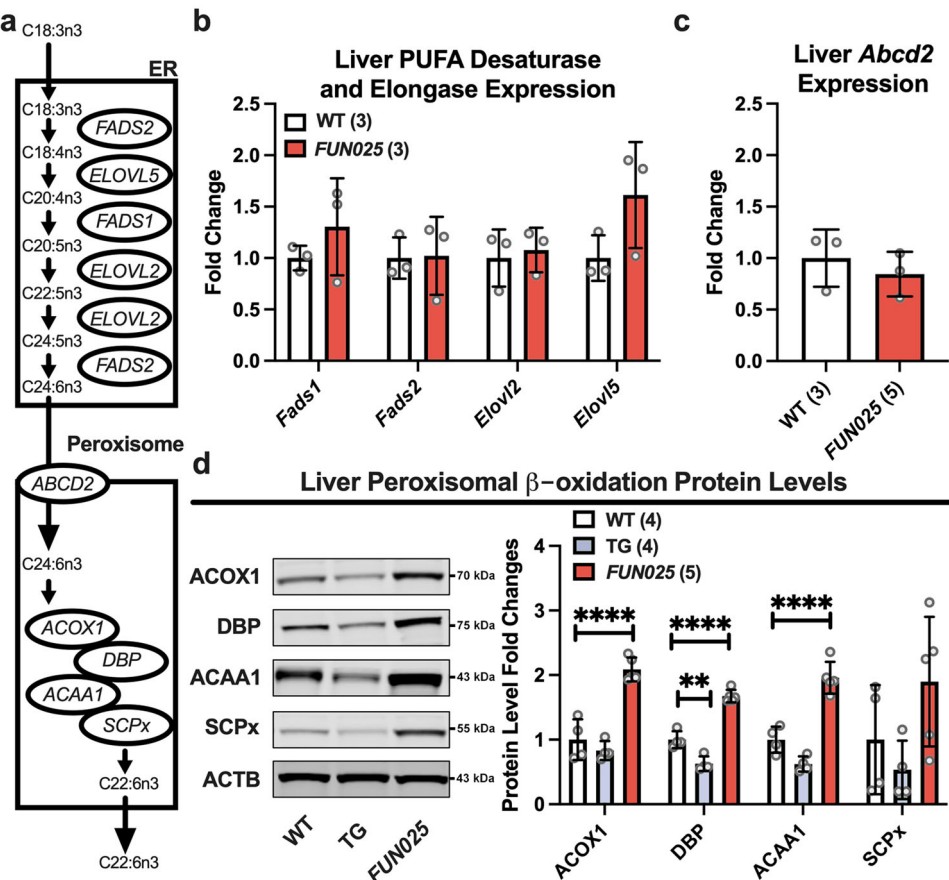

**Fig. 3 Peroxisomal beta-oxidation enzymes are increased in *Tmem135* mutant livers. a** Schematic of the Sprecher pathway of docosahexaenoic acid (C22:6n3) synthesis from alpha-linolenic acid (C18:3n3) through a series of elongation and desaturation steps in the endoplasmic reticulum (ER) that is finished within peroxisomes via their beta-oxidation machinery. **b** Gene expression analysis of the ER-localized *Fatty acid desaturase 1* (*Fads1*) and *2* (*Fads2*) and *Elongation of very-long-chain fatty acids-like 2* (*Elovl2*) and *5* (*Elovl5*) involved in the Sprecher pathway in the livers of three 2.5-month-old female WT and *Tmem135FUN025/FUN025* (*FUN025*) mice. **c** Gene expression analysis of *ATP binding cassette subfamily D member 2* (*Abcd2*) in the livers of three 2.5-month-old female WT and *Tmem135FUN025/FUN025* mice. *Ribosomal protein lateral stalk subunit P0* (*Rlplp0*) served as the housekeeping gene in these studies. **d** Western blot analysis of peroxisomal beta-oxidation enzymes including acyl-CoA oxidase 1 (ACOX1), D-bifunctional protein (DBP), acetyl-Coenzyme A acyltransferase 1 (ACAA1), and sterol carrier protein x (SCPx) using liver lysates from 2.5-month-old WT, *Tmem135* TG (TG), and *Tmem135FUN025/FUN025* mice. ACTB served as the loading control for these experiments. 4 WT (2 males/2 females), 4 *Tmem135* TG (2 males/2 females), and 5 *Tmem135FUN025/FUN025* (3 males/2 females) were used in these experiments. Asterisks (** and ****) indicates post hoc Tukey test for a $P < 0.01$ and $P < 0.0001$ significance following a significant difference detected by one-way ANOVA. Dots represent individual data points. The number in parentheses represents the N of independent mouse samples per genotype used in the experiment. The protein size next to the immunoblot images denotes the size of the immunoband measured for this analysis. These experiments were repeated twice to ensure reproducibility. Data are presented as mean ± SD.

*Lepob/ob* mice. We found that 3-month-old male and female *Tmem135FUN025/FUN025/Lepob/ob* mice had smaller body weights (Fig. 7a), liver weights (Fig. 7b), and gonadal fat pad weights (Fig. 7c) than age-matched *Lepob/ob* mice. Plasma cholesterol levels were also significantly reduced in male and female *Tmem135FUN025/FUN025/Lepob/ob* mice compared to *Lepob/ob* controls (Fig. 7d), but no changes in plasma triglycerides (Fig. 7e) and non-fasting glucose (Fig. 7f) were observed between these genotypes. We measured the amounts of plasma apolipoproteins in *Tmem135FUN025/FUN025/Lepob/ob* mice to determine if the decreased plasma cholesterol in this genotype resulted from altered lipoprotein concentrations. We found decreases in apolipoprotein B100 (APOB100) in the plasmas of *Tmem135FUN025/FUN025/Lepob/ob* male (Fig. 7g) and female mice (Supplementary Fig. 7). No significant change in plasma apolipoprotein B48 (APOB48) (Fig. 7g) and apolipoprotein A1 (APOA1) (Fig. 7h) were observed between these male genotypes. APOB100 is a marker for very-low-density lipoprotein (VLDL) and low-density lipoprotein (LDL), both of which are major carriers of cholesterol

from the liver to peripheral tissues[42]. Thus, the decreased plasma cholesterol in *Tmem135FUN025/FUN025/Lepob/ob* mice can be explained by their decreased plasma APOB100-containing lipoproteins. In summary, the *Tmem135* mutation reduces obesity and improves dyslipidemia in male and female *Lepob/ob* mice.

***Tmem135* mutation ameliorates the liver disease of leptin mutant mice**. *Lepob/ob* mice develop non-alcoholic fatty liver disease (NAFLD), a liver condition categorized by excessive fat accumulation that can progress to inflammation and fibrosis[43]. We examined the liver sections of 3-month-old *Tmem135FUN025/FUN025/Lepob/ob* and *Lepob/ob* mice and found that *Tmem135-FUN025/FUN025/Lepob/ob* mice had less severe NAFLD pathologies relative to *Lepob/ob* mice (Fig. 8a). We also observed less oil red o staining of liver sections from *Tmem135FUN025/FUN025/Lepob/ob* mice compared to *Lepob/ob* mice, indicating less neutral lipid accumulation in *Tmem135FUN025/FUN025/Lepob/ob* mice (Fig. 8a). We found no signs of a histological phenotype in the livers of *Tmem135FUN025/FUN025* mice (Fig. 8a). Next, we calculated

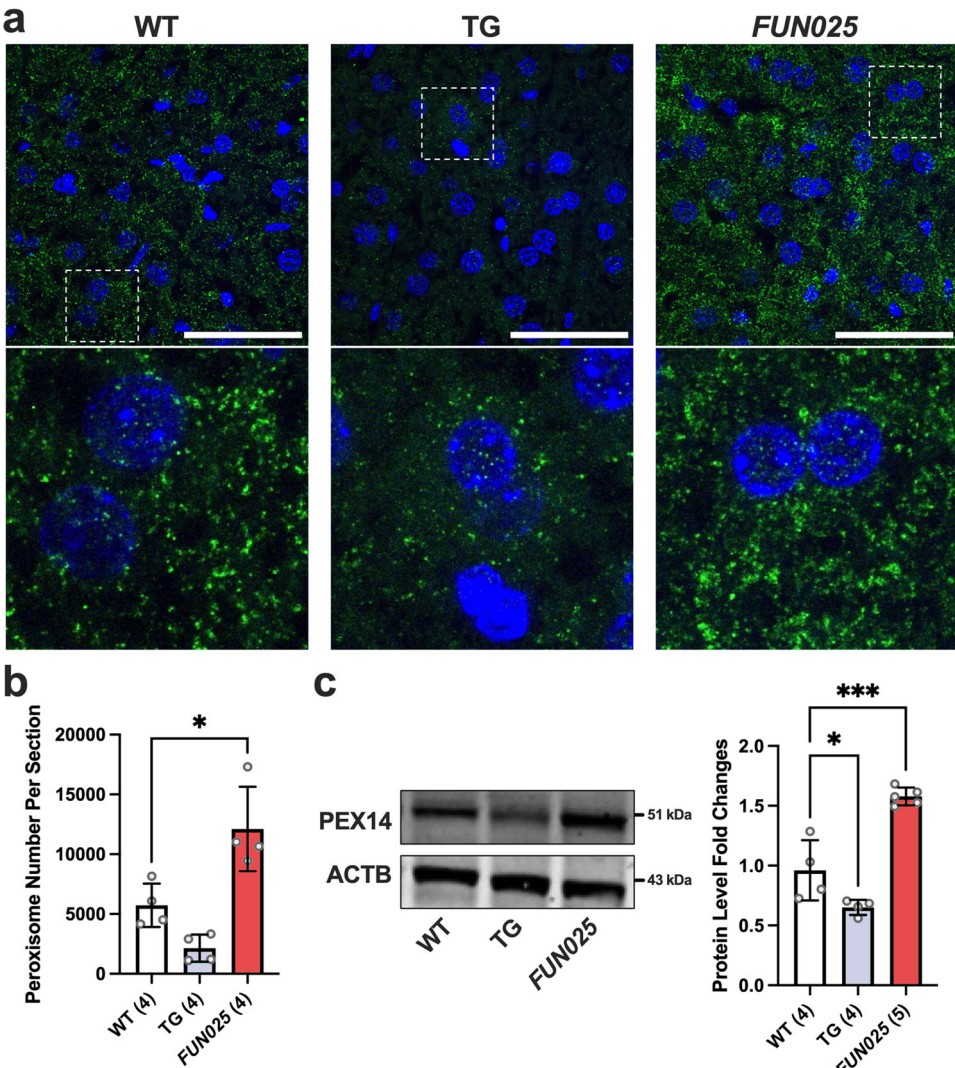

**Fig. 4 TMEM135 regulates the number of peroxisome biogenesis factor 14 (PEX14) positive peroxisomes in mice. a** Representative 100x immunohistochemical images of PEX14 labeled (green) and DAPI stained (blue) WT, *Tmem135* TG (TG), and *Tmem135*$^{FUN025/FUN025}$ (*FUN025*) livers. The white boxes in these images were expanded to highlight differences in PEX14-positive peroxisome staining between these genotypes. Scale bar for images = 50 microns. **b** Quantitation of PEX14-positive peroxisomes from the 100x images of WT (2 males/2 females), *Tmem135* TG (1 male/3 females), and *Tmem135*$^{FUN025/FUN025}$ (2 males/2 females) livers using the Analyze Particles function in ImageJ. **c** Western blot analysis of peroxisome biogenesis factor 14 (PEX14) using livers from 2.5-month-old WT (2 males/2 females), *Tmem135* TG (2 males/2 females), and *Tmem135*$^{FUN025/FUN025}$ (3 males/2 females) mice. ACTB served as the loading control for this Western blot experiments. Asterisks (* and ***) indicates post hoc Tukey test for a $P < 0.05$ and $P < 0.001$ significance following a significant difference detected by one-way ANOVA. Number in parentheses represent the $N$ of independent mouse samples per genotype used in the experiment. Dots represent individual data points. The protein size next to the immunoblot images denotes the size of the immunoband measured for this analysis. Data are presented as mean ± SD.

NAFLD scores based on the incidence of microvesicles, macrovacuoles, and hypertrophy. Both male and female *Tmem135*$^{FUN025/FUN025}$/*Lep*$^{ob/ob}$ mice had lower NAFLD scores than *Lep*$^{ob/ob}$ mice (Fig. 8b). To assess if these hepatic histological results equated to liver injury changes, we measured the activity of alanine transaminase (ALT), a liver enzyme that is leaked into the plasma after liver injury[44] and increased in *Lep*$^{ob/ob}$ plasmas[45]. Plasma ALT activities were significantly reduced in both male and female *Tmem135*$^{FUN025/FUN025}$/*Lep*$^{ob/ob}$ mice compared to *Lep*$^{ob/ob}$ mice (Fig. 8c). These results indicate that the *Tmem135* mutation ameliorates NAFLD in *Lep*$^{ob/ob}$ mice.

**_Tmem135_ mutation augments peroxisomal proteins in leptin mutant mice.** The absence of leptin leads to increased lipid synthesis and decreased lipid oxidation, setting the stage for

obesity, dyslipidemia, and NAFLD development in *Lep*$^{ob/ob}$ mice[46]. We speculate that the *Tmem135* mutation ameliorates these phenotypes in *Lep*$^{ob/ob}$ mice by increasing peroxisomal beta-oxidation. We sought to confirm if the *Tmem135* mutation caused lipid and peroxisomal changes in *Lep*$^{ob/ob}$ mice that we had observed in *Tmem135* mutant mice. We evaluated the lipid profiles of 3-month-old male *Lep*$^{ob/ob}$ and *Tmem135*$^{FUN025/FUN025}$/*Lep*$^{ob/ob}$ plasmas. Principal component analysis revealed a lipid profile in *Tmem135*$^{FUN025/FUN025}$/*Lep*$^{ob/ob}$ plasmas that was different from *Lep*$^{ob/ob}$ plasmas (Fig. 9a). There were many lipid species modified in *Tmem135*$^{FUN025/FUN025}$/*Lep*$^{ob/ob}$ plasmas (Supplementary Fig. 8, Supplementary Table 4) but lipids containing DHA were significantly decreased in *Tmem135*$^{FUN025/FUN025}$/*Lep*$^{ob/ob}$ plasmas relative to *Lep*$^{ob/ob}$ plasmas (Fig. 9b, Supplementary Table 5). We confirmed the increases of

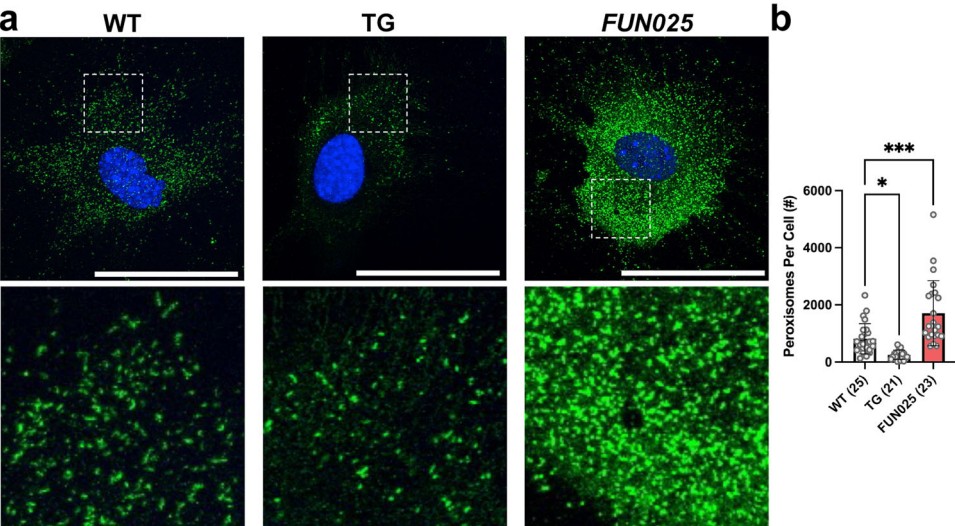

**Fig. 5 TMEM135 regulates peroxisome proliferation in vitro. a** Representative 60x immunohistochemical images of peroxisome biogenesis factor 14 (PEX14) labeled (green) and DAPI stained (blue) WT, *Tmem135* TG (TG), and *Tmem135*$^{FUN025/FUN025}$ (*FUN025*) fibroblasts. The white boxes in these images were expanded to show differences in the peroxisome number between these cells. Scale bar for images = 50 microns. **b** Quantitation of PEX14-positive peroxisomes from WT, *Tmem135* TG, and *Tmem135*$^{FUN025/FUN025}$ livers using the Analyze Particles function in ImageJ. Asterisks (* and ***) indicates post hoc Tukey test for a $P < 0.05$ and $P < 0.001$ significance following a significant difference detected by one-way ANOVA. Number in parentheses represent the *N* of individual fibroblasts per genotype assessed in this experiment. Dots represent individual data points. Data are presented as mean ± SD.

peroxisomal beta-oxidation enzymes ACOX1, DBP, and ACAA1 in male (Fig. 9c) and female (Supplementary Fig. 9a) *Tmem135*$^{FUN025/FUN025}$/*Lep*$^{ob/ob}$ liver lysates compared to gender-matched *Lep*$^{ob/ob}$ liver lysates. Additionally, peroxisome markers PEX14, PMP70, and CAT were significantly increased in male (Fig. 9d) and female (Supplementary Fig. 9b) *Tmem135*$^{FUN025/FUN025}$/*Lep*$^{ob/ob}$ liver lysates. This data verified that the *Tmem135* mutation reduces DHA-containing lipids but increases peroxisomal proteins in *Lep*$^{ob/ob}$ mice.

## Discussion

Maintaining lipid metabolism is an important task for cells to sustain their health. Here, we sought to identify the role of TMEM135 in lipid metabolism. We found a mutation in *Tmem135* significantly altered the lipid profiles of multiple murine tissues including the liver, retina, heart, and plasma. Remarkably, a major discovery from our lipidomics analysis was severely diminished concentrations of DHA-containing lipids in all the *Tmem135* mutant (*Tmem135*$^{FUN025/FUN025}$) tissues evaluated in this study. We confirmed the results from our lipidomic analysis by quantifying fatty acids and detecting substantial decreases of DHA in the livers, retinas, hearts, and plasmas from *Tmem135* mutant mice. DHA is an important modulator of disease[47]. For instance, DHA can be used to synthesize anti-inflammatory eicosanoids required for immune responses[30] and mediate fatty acid synthesis[48]. DHA is also an abundant component of photoreceptor membranes required for their fluidity and permeability[49]. The decreased DHA in *Tmem135* mutant mice may explain their global lipid profile changes and severe retinal degeneration[13,15,27] that phenocopies other mice with DHA deficiencies such as elongation of very-long-chain fatty acids-like 2 (*Elovl2*) mutant[50,51], major facilitator superfamily domain containing 2A (*Mfsd2a*) knockout[52–54], and adiponectin receptor 1 (*Adipor1*) knockout mice[55,56]. Altogether, this evidence supports the role of TMEM135 in cellular DHA homeostasis.

The origin of DHA in mice stems from the contributions of dietary intake and cellular production of DHA. Since there is no DHA present in the food available to the mice used in this study, the reduction of DHA in *Tmem135* mutant mice must originate from a defect in its endogenous production. However, we did not observe any decreases in the components required for DHA synthesis in the *Tmem135* mutant livers that could explain their decreased DHA concentrations. The remaining step that could be affected is the export of DHA synthesized in peroxisomes. While little is known about how DHA leaves the peroxisome, it has been postulated that there is a protein responsible for the export of DHA from the peroxisome[57]. Since TMEM135 is present on peroxisomal membranes[16–20,26], we hypothesize that TMEM135 exports DHA from peroxisomes for its use by cells (Fig. 10a). TMEM135 may either modulate the activity of the peroxisomal DHA exporter or function as the peroxisomal DHA exporter. We predict that the *Tmem135* mutation leads to the retention of DHA within the peroxisome, causing DHA to be degraded by peroxisomal beta-oxidation (Fig.10b). There lies the key difference between *Tmem135* mutant mice and other mouse models displaying DHA deficiencies from peroxisomal abnormalities. Mice with peroxisomal biogenesis defects that are unable to produce functional peroxisomes such as peroxisome biogenesis factor 2[58] and peroxisome biogenesis factor 5 knockout mice[59] have reduced DHA concentrations. Also, mice with peroxisomal beta-oxidation defects such as *Acox1*[60] and multifunctional protein 2 (also known as DBP) knockout mice[61] have decreased levels of DHA. However, these mice obviously lack peroxisome functions in general or peroxisomal beta-oxidation capacity, while *Tmem135* mutant mice retain them. These unique characteristics may underlie the increase in peroxisomes that occurs in *Tmem135* mutant mice as described below.

In addition to their role in DHA biogenesis through the Sprecher pathway, peroxisomes perform other unique anabolic and catabolic lipid metabolic functions that are critical for cellular homeostasis[8,62,63]. In this study, we found that the number of peroxisomes in murine livers and fibroblasts was negatively

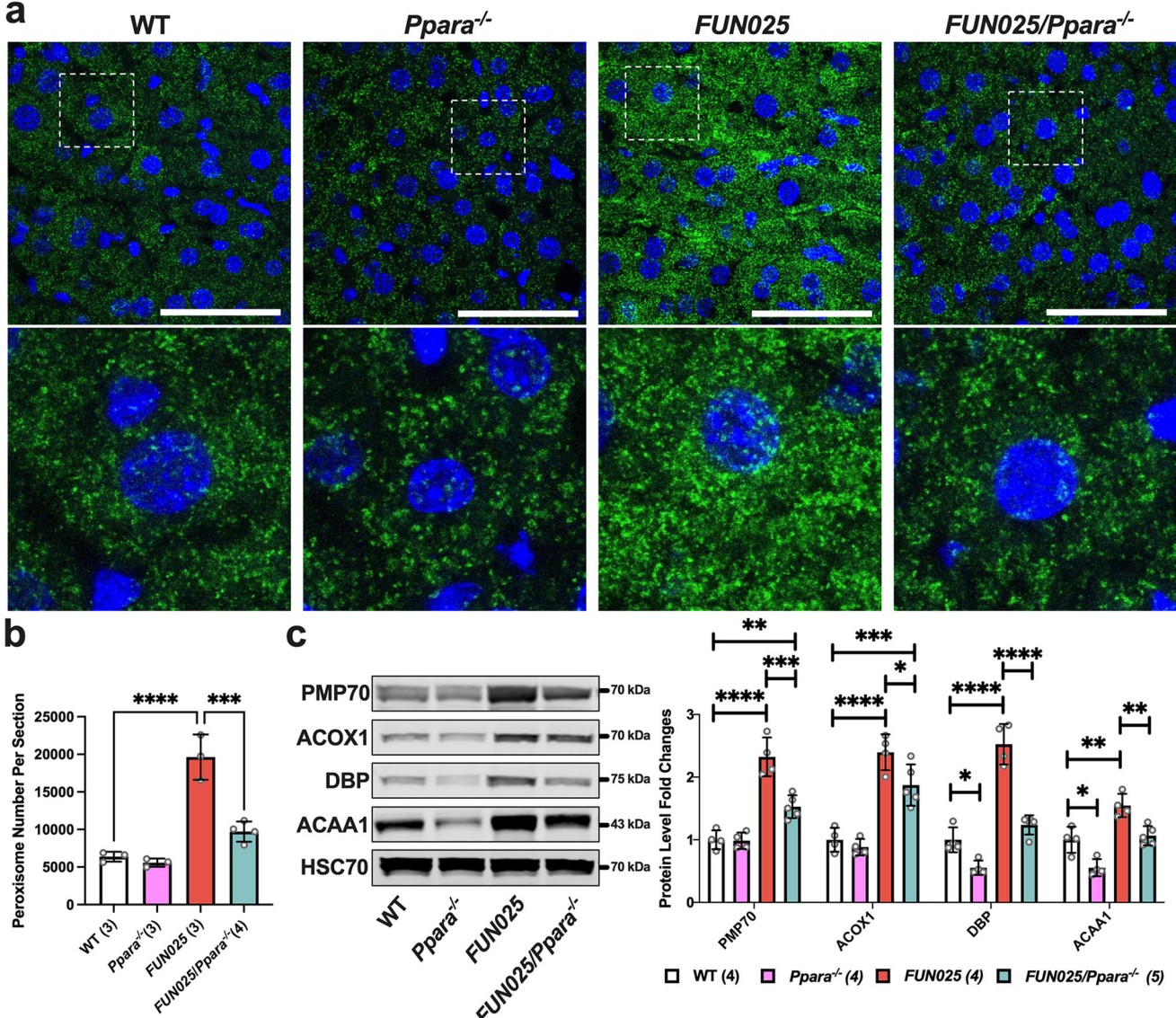

**Fig. 6 *Tmem135* mutation activates peroxisome proliferator-activated receptor alpha (PPARa) in the mouse liver. a** Representative 100x immunohistochemical images of PEX14 labeled (green) and DAPI stained (blue) 3-month-old WT, *Ppara*$^{-/-}$, *Tmem135*$^{FUN025/FUN025}$ (FUN025), and *Tmem135*$^{FUN025/FUN025}$/*Ppara*$^{-/-}$ (FUN025/*Ppara*$^{-/-}$) livers. The white boxes in these images were expanded to highlight differences of the PEX14-positive peroxisome staining between these genotypes. Scale bar for images = 50 microns. **b** Quantitation of PEX14-positive peroxisomes from the 100x images of WT (1 male/ 2 females), *Ppara*$^{-/-}$ (1 male/2 females), *Tmem135*$^{FUN025/FUN025}$ (1 male/2 females), and *Tmem135*$^{FUN025/FUN025}$/*Ppara*$^{-/-}$ (1 male/ 3 females) livers using the Analyze Particles function in ImageJ. **c** Western blot analysis of peroxisomal proteins in the livers of 3-month-old WT (3 males/1 female), *Ppara*$^{-/-}$ (2 males/2 females), *Tmem135*$^{FUN025/FUN025}$ (2 males/2 females), and *Tmem135*$^{FUN025/FUN025}$/*Ppara*$^{-/-}$ (3 males/2 females) mice. HSC70 served as the loading control for these experiments. Asterisks (*, **, ***, and ****) indicates post hoc Tukey test for a $P < 0.05$, $P < 0.01$, $P < 0.001$, and $P < 0.0001$ significance following a significant difference detected by one-way ANOVA. PMP70, peroxisomal membrane protein 70. ACOX1, acyl-CoA oxidase 1. DBP, D-bifunctional protein. ACAA1, acetyl-Coenzyme A acyltransferase 1. Dots represent individual data points. Number in parentheses represents *N* of independent mouse samples per genotype used in the experiment. The protein size next to the immunoblot images denotes the size of the immunoband measured for this analysis. Data are presented as mean ± SD.

correlated with the amount of functional TMEM135. The mutation in *Tmem135* augments the number of peroxisomes while over-expression of *Tmem135* reduces the number of peroxisomes, indicating an important role of TMEM135 function in maintaining the number of peroxisomes. Additionally, the increased number of peroxisomes due to the *Tmem135* mutation is consistent with augmented peroxisomal beta-oxidation enzymes including ACOX1, DBP, and ACAA1 in the *Tmem135* mutant livers. We posit that the increased beta-oxidation of DHA retained in peroxisomes of *Tmem135* mutant mice may alter the concentrations of peroxisome-derived metabolites known to drive

peroxisomal biogenesis or division potentially through PPAR signaling (Fig. 10c)[35,64]. Consistent with this notion, we found that peroxisomes and their protein contents were decreased in *Tmem135* mutant livers with the genetic ablation of *Ppara*, demonstrating that PPAR signaling is at least partly mediating peroxisome levels in the *Tmem135* mutant liver. A ligand that could be activating PPAR signaling in the *Tmem135* mutant liver is ether phosphatidylethanolamines (EtherPEs)[10], which are produced by peroxisomes[10] and are higher in *Tmem135* mutant tissues (Supplementary Table 1). In fact, EtherPE16:1e_18:1 and EtherPE16:1e_18:2 are commonly increased across *Tmem135*

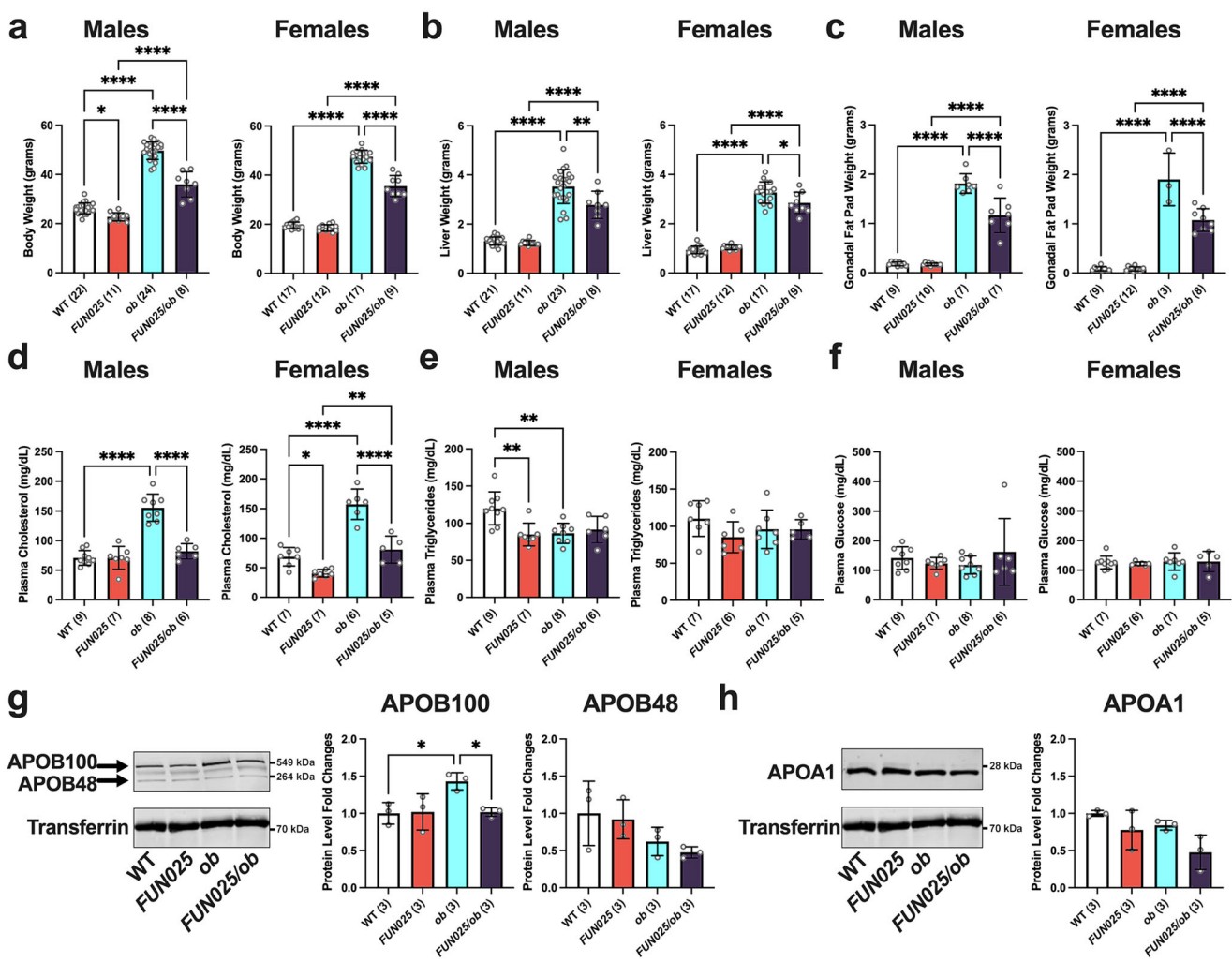

**Fig. 7 *Tmem135* mutation reduces leptin mutation-induced obesity and dyslipidemia. a** Body, **b** liver, and **c** gonadal fat pad weights as well as **d** plasma cholesterol, **e** plasma triglyceride, and **f** plasma non-fasting glucose levels of 3-month-old WT, *Tmem135^{FUN025/FUN025}* (*FUN025*), *Lep^{ob/ob}* (*ob*), and *Tmem135^{FUN025/FUN025}/Lep^{ob/ob}* (*FUN025/ob*) male and female mice. Western blot analysis of plasmas from 3-month-old WT, *Tmem135^{FUN025/FUN025}*, *Lep^{ob/ob}*, and *Tmem135^{FUN025/FUN025}/Lep^{ob/ob}* male mice for **g** apolipoprotein B100 (APOB100) and B48 (APOB48) and **h** apolipoprotein A1 (APOA1). Transferrin served as a loading control for these experiments. Number in parentheses represent the *N* of independent mice or mouse samples per genotype used in the experiment. Dots represent individual data points. The protein size next to the immunoblot images denotes the size of the immunoband measured for this analysis. Asterisks (*, **, and ****) indicate a *P* < 0.05, *P* < 0.01, and *P* < 0.0001 significance by post hoc Tukey test following a significant difference detected by one-way ANOVA. Data are presented as mean ± SD.

mutant tissues (Supplementary Tables 6–9). Further investigation is necessary to identify the exact ligands that modify the number of peroxisomes in *Tmem135* mutant mice as well as to determine whether these ligands are responsible for peroxisome proliferation observed in mouse models of peroxisomal beta-oxidation deficiencies[65–67].

Our study also uncovered the effect of the *Tmem135* mutation on the phenotypes of leptin mutant mice. The leptin mutant mouse is a well-characterized mouse model of dysregulated lipid metabolism often used to evaluate key pathways involved in metabolic diseases[43,68]. We found leptin mutant mice that are homozygous for the *Tmem135* mutation have reduced body weight, smaller gonadal fat pads, lower plasma cholesterol, and decreased plasma APOB compared to leptin mutant mice. We also observed that the *Tmem135* mutation ameliorates fatty liver disease in leptin mutant mice. We postulate that the protection provided by the *Tmem135* mutation on leptin mutant phenotypes can be explained by enhanced peroxisomal beta-oxidation from the activation of PPARa signaling due to the *Tmem135* mutation

(Fig. 10c). We support this idea through a study evaluating wildtype and *Tmem135* mutant mice in metabolic phenotyping cages that found the *Tmem135* mutation increased fatty acid oxidation in mice (Supplementary Fig. 10). It was previously shown that the level of peroxisomal beta-oxidation can affect the phenotypes of leptin mutant mice. For example, leptin mutant mice treated with fenofibrate, a PPARa agonist[69], decreased body weight, reduced plasma cholesterol, lessened fat accumulation, and improved their fatty liver phenotypes[41]. In contrast, mice with the genetic ablation of *Ppara* on the leptin mutant background showed worsened obesity, increased fat accumulation, and more severe hepatic steatosis compared to leptin mutant mice[40]. In support of increased PPARa activation in the double *Tmem135* and leptin mutant mice, we confirmed the higher concentrations of hepatic peroxisomal markers and beta-oxidation enzymes in these mice compared to leptin mutant mice.

In summary, we investigated a role of TMEM135 in lipid metabolism. We found TMEM135 has a critical role in maintaining cellular DHA levels in mice that is abolished by a

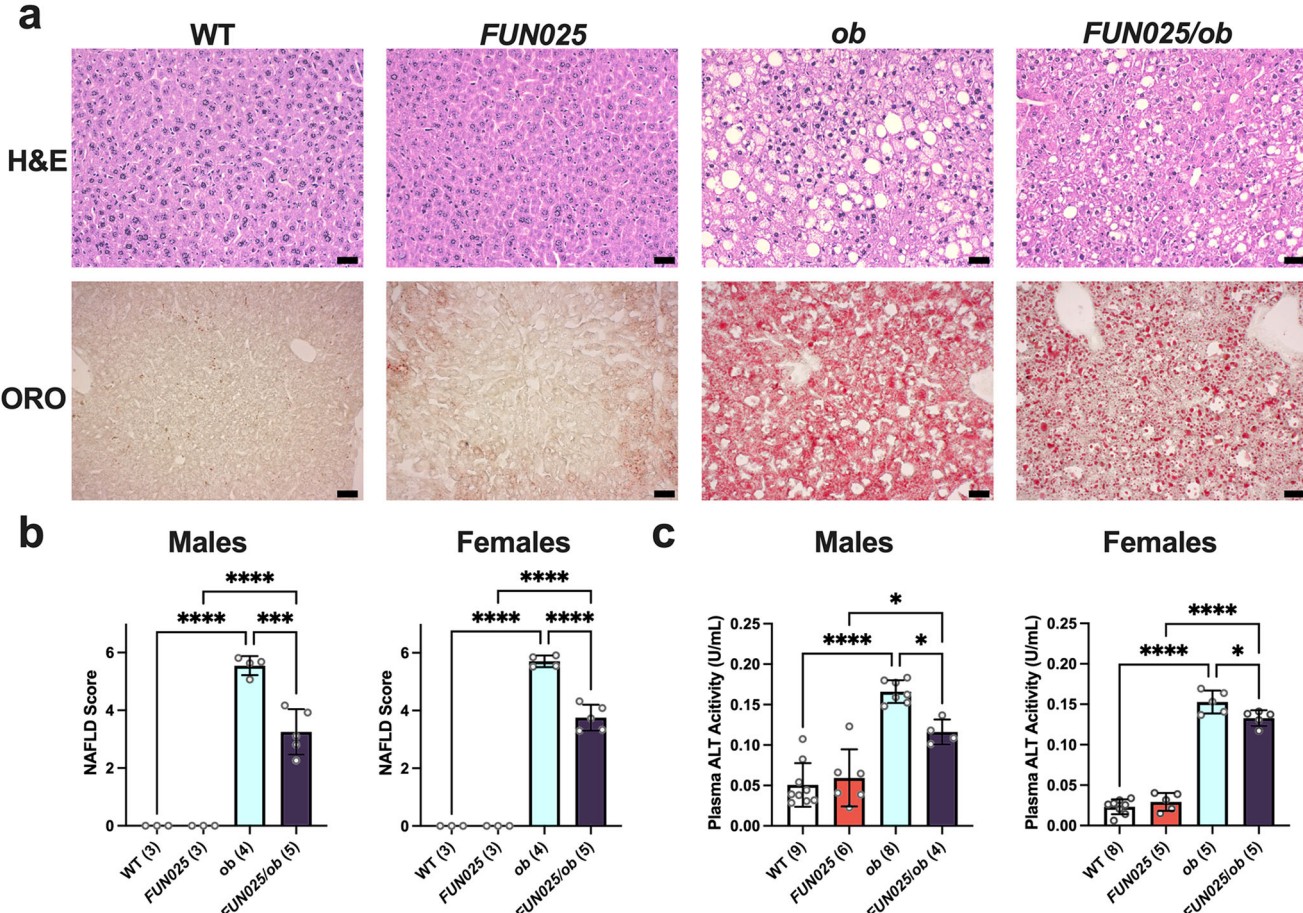

**Fig. 8 *Tmem135* mutation reduces leptin mutation-induced liver phenotype. a** Representative 20x images of hematoxylin and eosin (H&E) and oil red o (ORO) stained liver sections from 3-month-old male WT, *Tmem135*[FUN025/FUN025] (FUN025), *Lep*[ob/ob] (*ob*), and *Tmem135*[FUN025/FUN025]/*Lep*[ob/ob] (FUN025/ *ob*) mice. Scale Bar = 100 microns. **b** Non-alcoholic fatty liver disease (NAFLD) severity scores. **c** Plasma ALT activity. Number in parentheses represent the *N* of independent mouse samples per genotype used in the experiment. Dots represent individual data points. Asterisks (*, ***, and ****) indicates a $P < 0.05$, $P < 0.01$, $P < 0.001$, and $P < 0.0001$ significance by post hoc Tukey test following a significant difference detected by one-way ANOVA. Data are presented as mean ± SD.

mutation in the murine *Tmem135* gene. Since we did not observe any decreases in the components involved in cellular DHA synthesis, we hypothesized that the *Tmem135* mutation prevents the export of DHA from peroxisomes, leading to the retention and degradation of DHA within peroxisomes in the *Tmem135* mutant tissues. We also discovered TMEM135 is important for maintaining peroxisomal number and function through the regulation of PPARa signaling. Lastly, we established that the *Tmem135* mutation is protective against leptin mutant-induced mouse phenotypes that can be explained by the increased peroxisomal beta-oxidation induced by the *Tmem135* mutation, Together, we conclude TMEM135 has a role in maintaining lipid homeostasis in mice through its function in peroxisomes.

It is important to note that there were limitations of this study concerning the origin of protection of the *Tmem135* mutation on leptin mutant mouse phenotypes. For example, the impact of the *Tmem135* mutation on other organelles known to harbor TMEM135 such as mitochondria[13,20] and lipid droplets[20,21] could possibly explain the decreased severity of leptin mutant mouse phenotypes. We detected changes in mitochondrial proteins carnitine palmitoyltransferase 1a (CPT1a) (Supplementary Fig. 11) and carnitine-acylcarnitine translocase (CACT) (Supplementary Fig. 11) but no differences in perilipin 2 (PLIN2) (Supplementary Fig. 12), the main protein constituent of hepatic lipid droplets[70], in *Tmem135* mutant livers. Interestingly, hepatic

CPT1a protein levels were decreased in *Tmem135* mutant livers (Supplementary Fig. 11). *Cpt1a* is a known target of PPARa signaling[71] that we found to be activated in *Tmem135* mutant livers. The decreased CPT1a could result from a PPARa-independent regulatory mechanism involving epigenetic and posttranslational modifications that would affect its protein levels[71]. Alternatively, the decreased CPT1a could reflect an impaired function of TMEM135 on mitochondrial membranes since the *Tmem135* mutation has profound effects on mitochondrial homeostasis[13–15,22,23]. CPT1a is the rate-limiting step for mitochondrial fatty acid oxidation that converts acyl-CoA esters to acylcarnitines for their import through the mitochondrial outer membrane[71]. It has been shown that liver-specific *Cpt1a* knockout mice were protected against diet-induced weight gain[72]. Thus, decreased hepatic CPT1a concentrations in *Tmem135* mutant mice may explain the smaller body weight of these mice and leptin double mutant mice compared to leptin mutant controls. Future studies are warranted to investigate whether the mitochondrial dysfunction caused by the *Tmem135* mutation is from its role on peroxisomes or mitochondria as this may shed insight into the mechanisms underlying its protection against the leptin mutant mouse phenotypes.

It is also possible that the protection of the *Tmem135* mutation on leptin mutant mouse phenotypes is from its effect on another tissue. We have shown in our study that multiple tissues in

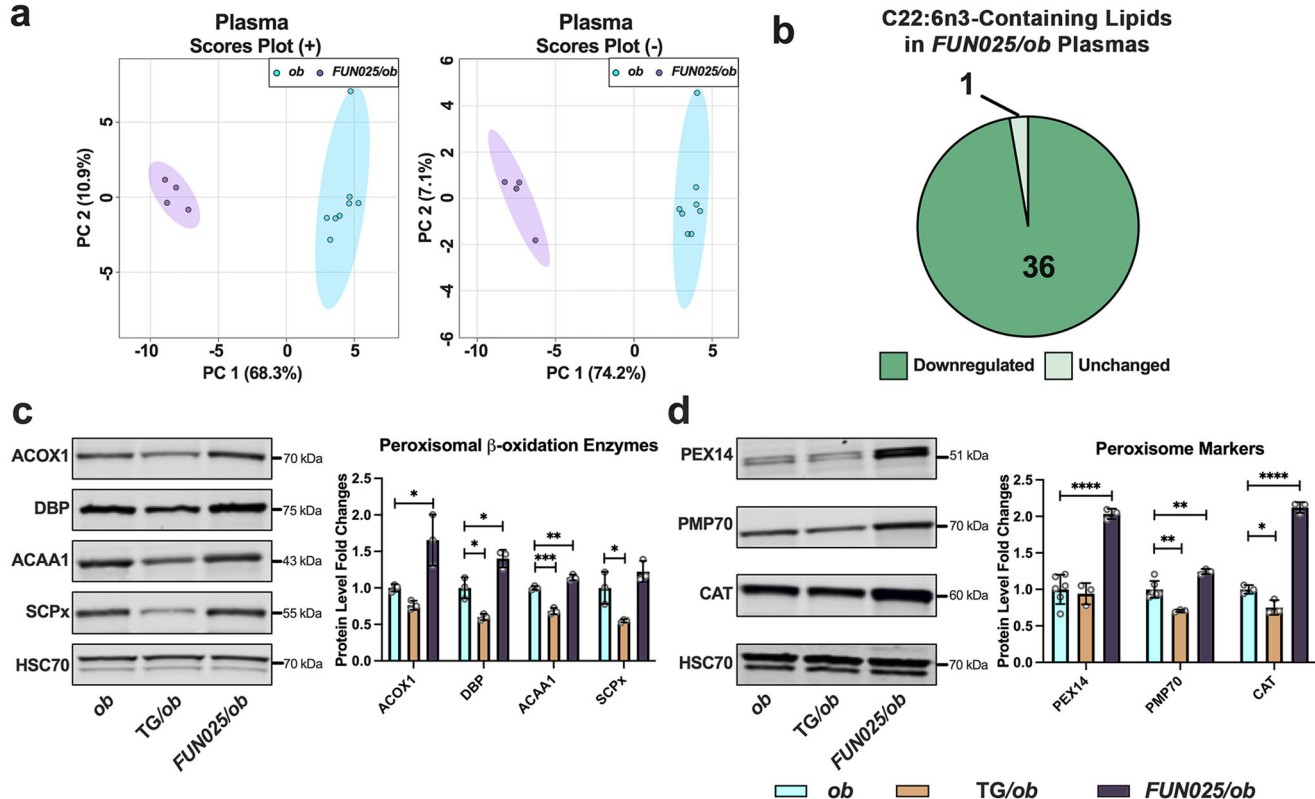

**Fig. 9 *Tmem135* mutation decreases docosahexaenoic polyunsaturated fatty acid-containing lipids and increases peroxisomal proteins in leptin mutant mice. a** Principal component analysis of lipids detected in the positive (+) ion mode and negative (–) ion mode of 3-month-old male $Lep^{ob/ob}$ ($ob$) ($N = 8$) and $Tmem135^{FUN025/FUN025}/Lep^{ob/ob}$ ($FUN025/ob$) ($N = 4$) plasmas based on log10 concentrations. **b** Pie graph of docosahexaenoic acid (DHA, C22:6n3)-containing lipids that were significantly downregulated (medium green) and unchanged (light green) in $Tmem135^{FUN025/FUN025}/Lep^{ob/ob}$ plasmas compared to $Lep^{ob/ob}$. Numbers denote the total of lipids within the category. All altered lipid species including DHA-containing lipids can be found in Supplementary Table 10. Significance was determined by two-way Student's t test ($P < 0.05$). **c** Western blot analysis of peroxisomal beta-oxidation enzymes including acyl-CoA oxidase 1 (ACOX1), D-bifunctional protein (DBP), acetyl-Coenzyme A acyltransferase 1 (ACAA1), and sterol carrier protein x (SCPx) using liver lysates from 3-month-old male $Lep^{ob/ob}$ ($N = 3$), $Tmem135$ TG/$Lep^{ob/ob}$ (TG/ob) ($N = 3$) and $Tmem135^{FUN025/FUN025}/Lep^{ob/ob}$ ($N = 3$) mice. **d** Western blot analysis of peroxisome biogenesis factor 14 (PEX14), peroxisome membrane protein 70 (PMP70), and catalase (CAT) using livers from 3-month-old male $Lep^{ob/ob}$ ($N = 3–6$), $Tmem135$ TG/$Lep^{ob/ob}$ (TG/ob) ($N = 3$) and $Tmem135^{FUN025/FUN025}/Lep^{ob/ob}$ ($N = 3$) mice. HSC70 served as the loading control for these experiments. Asterisks (*, **, ***, and ****) indicates post hoc Tukey test for a $P < 0.05$, $P < 0.01$, $P < 0.001$, and $P < 0.0001$ significance following a significant difference detected by one-way ANOVA. Dots represent individual data points. The protein size next to the immunoblot images denotes the size of the immunoband measured for this analysis. Data are presented as mean ± SD.

*Tmem135* mutant mice exhibit similar changes in DHA concentrations and peroxisomal proteins. Therefore, additional experiments are required to determine the contribution of specific tissues towards the protection of the *Tmem135* mutation on the phenotypes of leptin mutant mice.

## Methods

**Mice**. Tg(CAG-*Tmem135*)#Aike (*Tmem135* TG) mice congenic on the C57BL/6J background and $Tmem135^{FUN025/FUN025}$ mice were used in this study[13–15]. B6;129S4-$Ppara^{tm1Gonz}$/J ($Ppara^{–/–}$) (Stock #008154) and B6.Cg-$Lep^{ob}$/J ($Lep^{ob}$) (Stock #000632) and were obtained from The Jackson Laboratory (Bar Harbor, ME) and bred in the animal facility at the University of Wisconsin-Madison. C57BL/6J mice served as WT controls for this study. All mice were fed a global soy protein-free extruded rodent diet (#2016, Envigo, Madison, WI) and housed in the Medical Sciences Center Vivarium at the University of Wisconsin-Madison. Both males and females were used in this study. All numbers of mice used in experiments are provided within the figures and their legends. All mouse procedures were performed in accordance with the protocols approved by the Animal Care and Use Committee at the University of Wisconsin-Madison. We adhere to the Animal Research: Reporting of In Vivo Experiments (ARRIVE) guidelines in reporting our animal research.

**Sample preparation for lipidomics analysis**. Livers, retinas (neural retinas and eyecups), hearts, and plasmas were collected and stored at −80 °C prior to lipid extraction. Prior to extraction, all solutions were pre-chilled on ice. Samples were

transported and cut on a dry ice-cooled stainless-steel plate and weighed on a Mettler (XSR205) analytical balance to the nearest hundredth of a milligram. This weight would be used to normalize samples for analysis. Fifty microliter aliquots of plasma from each mouse were used in this study. No normalization occurred with the plasma samples since the same volume for each sample was used in this study. Tissues were then placed into Qiagen PowerBead tubes (P/N 13113-50) for homogenization and extraction. Lipids were extracted in a solution of 250 μL PBS, 225 μL methanol containing internal standards (Avanti SPLASH LipidoMix (Lot#3307-07) at 10 μL per sample), and 750 μL MTBE (methyl tert-butyl ether)[73]. The sample was homogenized in three 30-second cycles alternated with a 5-minute rest on ice using a Qiagen TissueLyzer II operated at 30 Hz. The final rest on ice was 15 minutes. After centrifugation at 16,000 × g for 5 minutes at 4 °C, 500 μL of the upper phase was collected in 1.5 ml centrifuge tubes and evaporated to dryness in a Savant speedvac concentrator. Lipid samples were reconstituted in 150 μL of isopropanol. A process blank was prepared in parallel with the tissue samples during extraction and analyzed concurrently with the samples. A pooled sample, used for lipid identification and quality control, was prepared by taking equal volumes from each sample of a given tissue after final resuspension in IPA.

**LC-MS methods for lipidomics**. Sample analysis was done at different dilution factors for different tissues and ionization polarities. For each injection samples were diluted in isopropanol by adding a given sample volume to isopropanol in an LC-MS vial with deactivated glass insert (Agilent P/N 5182-0554 and 5183-2086), vortexed to mix, and spun at low speed to collect the liquid prior to placing in the autosampler for analysis. Lipid extracts were separated on a Waters Acquity UPLC BEH C18 1.7 μm 2.1 × 100 mm column coupled in tandem with a Waters Acquity UPLC BEH C18 1.7 μm VanGaurd pre-column 2.1 × 5 mm and maintained at

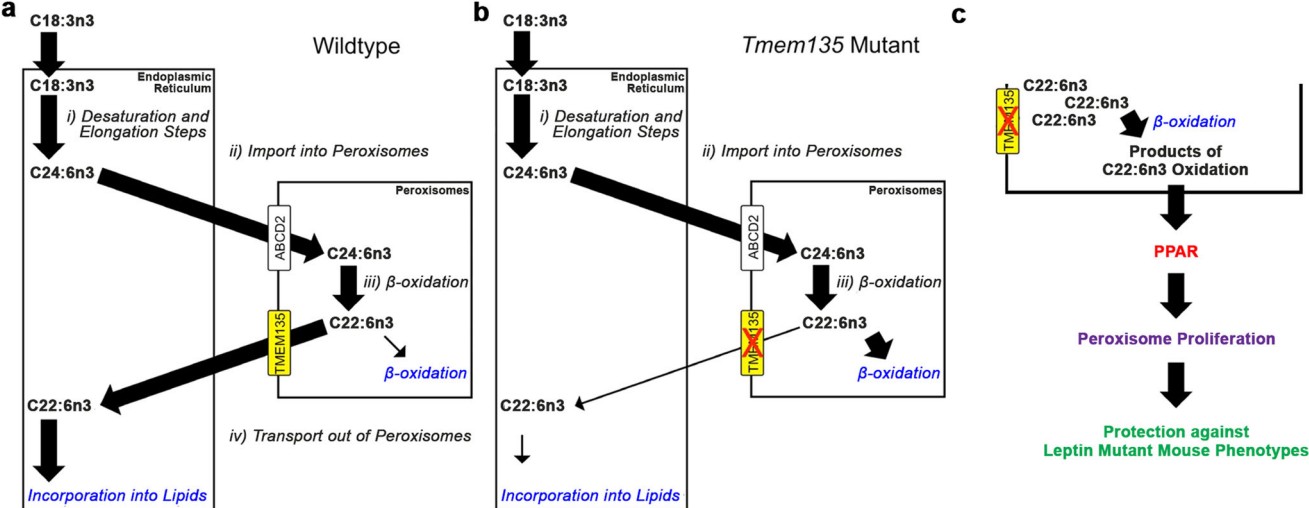

**Fig. 10 Schematic of proposed TMEM135 function. a** Cells depend on the interaction between the endoplasmic reticulum (ER) and peroxisomes to generate docosahexaenoic acid (C22:6n3). (i) The ER synthesizes C24:6n3 from C18:3n3 through a process of sequential desaturation and elongation steps. (ii) The ER transfers C24:6n3 to the cytosol for uptake by peroxisomes through ABCD2. (iii) One round of peroxisomal beta-oxidation converts C22:6n3 from C24:6n3. (iv) C22:6n3 leaves the peroxisome and migrates back to the ER for lipid incorporation. Since we observed reductions in DHA and no decreases of the components involved in steps i through iii in *Tmem135* mutant mice, we hypothesized TMEM135 functions by exporting C22:6n3 from the peroxisome. **b** The consequence of the *Tmem135* mutation may direct C22:6n3 away from the ER and toward peroxisomal beta-oxidation for its catabolism, thus accounting for the reduction of DHA in these mice. **c** The increased peroxisomal beta-oxidation of DHA within peroxisomes of *Tmem135* mutant mice may generate or alter ligands required for the activation of peroxisome proliferator-activated receptors (PPARs). The activation of PPARs can lead to the augmentation of peroxisomes and their beta-oxidation enzymes as we observed in *Tmem135* mutant mice. We hypothesized that the increased peroxisomes and their beta-oxidation enzymes due to the *Tmem135* mutation may confer protection against the development of leptin mutant mouse phenotypes.

50 °C. These columns in turn were connected to an Agilent HiP 1290 Multi-sampler, Agilent 1290 Infinity II binary pump, and column compartment connected to an Agilent 6546 Accurate Mass Q-TOF dual ESI mass spectrometer. For positive ion mode, the source gas temperature was set to 250 °C, with a gas flow of 12 L/min and a nebulizer pressure of 35 psig. VCap voltage was set at 4000 V, fragmentor at 145 V, skimmer at 45 V, and Octopole RF peak at 750 V. For negative ion mode, the source gas temperature was set to 350 °C, with a drying gas flow of 12 L/min and a nebulizer pressure of 25 psig. VCap voltage is set at 5000 V, fragmentor at 200 V, skimmer at 45 V, and Octopole RF peak at 750 V. Reference masses in positive mode (*m/z* 121.0509 and 922.0098) and negative mode (*m/z* 966.0007, and 112.9856) were delivered to the second emitter in the dual ESI source by isocratic pump at 15 uL/min. Samples were analyzed in a randomized order in both positive and negative ionization modes in separate experiments acquiring with the scan range m/z 100–1500. Mobile phase A consisted of acetonitrile:water (60:40 v/v) containing 10 mM ammonium formate and 0.1% formic acid, and mobile phase B consists of isopropanol: acetonitrile:water (90:9:1 v/v) containing 10 mM ammonium formate and 0.1% formic acid. The chromatography gradient for both positive and negative modes started at 15% mobile phase B then increases to 30% B over 2.4 min, then increased to 48% B from 2.4–3.0 min, then increased to 82% B from 3–13.2 min, then increased to 99% B from 13.2–13.8 min where it was held until 15.4 min and then returned to the initial conditions and equilibrated for 4 min. Flow was 0.5 mL/min throughout the gradient. Injection volumes were 2 μL for positive mode and 5 μL for negative mode MS[1] acquisitions, and 4 μL for positive mode, and 7 μL for negative mode MS/MS acquisitions. Tandem mass spectrometry (MS/MS) was conducted using the same LC gradient as above with isolation width set to "narrow" (~1.3 m/z) and collision energy of 25 V. MS/MS data were collected on the pooled sample per tissue in iterative mode, in which the sample is analyzed five times with different precursors selected upon each injection. This permits access to lower-abundance lipid species in a background of highly abundant lipids.

**Data analysis.** Pooled MS/MS and individual sample MS data were analyzed using a combination of Agilent and web-based applications. Lipid identification was achieved from the pooled MS/MS data via Lipid Annotator (Agilent). This software utilizes accurate mass m/z values for the intact lipid and experimental and theoretical fragment ion m/z values to assign lipid class and alkyl chain identities where possible. In some cases, specific alkyl chain identities cannot be assigned, but the sum composition of those alkyl chains (i.e., carbon number and degree of unsaturation) can be determined and that is reported instead. The output from Lipid Annotator is a database of lipid species with m/z values and retention times for each. This process was performed independently for positive and negative ion modes. Quantitation of lipids in individual samples used MS data alone. In this

analysis, data for each sample was collected separately. The abundance of each lipid in each sample was tabulated using the Profinder application (Agilent). In this analysis, the database of lipid identity, m/z values (for protonated, deprotonated, ammonium or sodium adducted, or formate adducted, depending on ionization mode), and retention time was used to extract an ion chromatogram for each lipid species. These chromatograms were then integrated to produce an area for that lipid in each sample. These abundance values were then exported as a .csv file for statistical analysis using the web-based tool MetaboAnalyst 5.0[74]. This software was used for principal component analysis and statistical testing. All significantly changed lipids in *Tmem135* mutant tissues identified in our study are provided in Supplementary Tables 6–10. All lipidomics data from this study is available on the Dryad public database (accession code: doi:10.5061/dryad.vx0k6djvm).

**Fatty acid composition analysis.** Total lipids from retina and plasma (100 μL) were extracted following the method of Bligh and Dyer[75] with modifications[76]. Total lipids from heart and liver were extracted following the method of Folch et al.[77]. To each lipid extract were added 25 nanomoles of 15:0 and 17:0 as internal standards. The lipid extracts were subjected to acid hydrolysis/methanolysis by heating at 100 °C for 2 h in 16% v/v concentrated HCl in methanol to generate fatty acid methyl esters (FAMEs). FAMEs were extracted into hexane and purified via TLC[78]. FAMEs were quantified using an Agilent Technologies 7890B gas chromatograph with flame ionization detector[79]. Data are represented as the relative mole percent of each fatty acid.

**Quantitative PCR.** Livers were collected from mice and kept at −80 °C degrees. RNA was extracted using a RNeasy lipid tissue mini kit (#74804, Qiagen, Germantown, MD) according to the manufacturer's instructions. RNA concentrations were determined using the Nanodrop 2000 UV-Vis Spectrophotometer (Thermo Fisher Scientific, Waltham, MA). One microgram of RNA was used to make cDNA using an Oligo d(T)18 primer (#S1316S, NEB, Ipswich, MA) and the ProtoScript II Reverse Transcriptase (#M0368L, NEB, Ipswich, MA) following the manufacturer's protocol. Expression of each gene was assessed in triplicate reactions in the Roche Lightcycler 480 system using cDNA, 200 nmol/L of each primer, and Lightcycler 480 SYBR Green I Supermix (#507203180, Roche). Relative mRNA expression was normalized to the ribosomal protein lateral stalk subunit P0 (*Rlpl0*) using the quantitative $2^{-\Delta\Delta CT}$ method[80]. Primer sequences used in this study can be found in Supplementary Table 11.

**Western blot analysis.** Tissues were isolated from mice and stored at −80 °C. Liver and heart lysates were made with an Ultra-Turrax T8 tissue homogenizer in RIPA buffer (#P189901, Thermo Fisher Scientific, Waltham, MA) containing

protease inhibitors (#11836170001, Thermo Fisher Scientific, Waltham, MA), respectively. Neural retina and eyecup (RPE/choroid/sclera) samples were homogenized using a Bel-Art Homogenizer system motor with RIPA buffer containing protease inhibitors. Protein concentrations were quantified using a BCA Protein Assay Kit (#P123228, Thermo Fisher Scientific, Waltham, MA). Equal protein amounts were aliquoted, reduced using XT Reducing Agent (#1610792, Biorad, Hercules, CA) for seven minutes at 105 °C, and loaded onto 10% Bis-Tris Criterion XT gels (#3450112, Biorad, Hercules, CA) in MOPS buffer (#1610788, Biorad, Hercules, CA) and transferred to nitrocellulose membranes (#102673-324, Biorad, Hercules, CA). Membranes were blocked with milk and probed overnight with their respective primary antibody at 4 °C. For Westerns using plasma samples, equal volumes of each sample were loaded in the gel but were not reduced with XT Reducing Agent. Also, these blots were blocked with bovine serum albumin diluted in PBS.

The primary antibodies and their dilutions used in this study can be found in Supplementary Table 12. Blots were washed with TBST buffer the next day and incubated with their corresponding secondary antibody. Secondary antibodies used in this study included donkey anti-rabbit IgG 680RD (#926-68073, LI-COR), donkey anti-rabbit IgG 800CW (#926-32213, LI-COR), donkey anti-guinea pig IgG 800CW (#925-32411, LI-COR), donkey anti-goat IgG 680RD (#926-68074, LI-COR), goat anti-mouse IgG1 800CW (#926-32350, LI-COR), goat anti-mouse IgG2a 800CW (#926-32351, LI-COR), and goat anti-mouse IgM 800CW (#925-32280, LI-COR). Blots were washed again with TBST and imaged using the Odyssey Imaging System (LI-COR Biosciences, Lincoln, NE) and analyzed using NIH's ImageJ (Besthesda, MD). Blots were stripped with Newblot Stripping Buffer (LI-COR Biosciences, Lincoln, NE) according to the manufacturer's protocol and reprobed with another primary antibody in this study. All immunobands were normalized to the loading control on their respective immunoblot.

**Immunohistochemistry for peroxisomal markers**. Tissues were fixed in 4% paraformaldehyde overnight at 4 °C, processed for cryoprotection, and embedded in Tissue-Tek O.C.T. Compound. Sections were cut at 8-micron thickness on a cryostat. We labeled peroxisomes with rabbit anti-PEX14 (#10594-1-AP, Proteintech, 1:200 dilution) or anti-PMP70 (#ab3421, Abcam, 1:200 dilution) and stained nuclei with 4′,6-Diamidine-2′-phenylindole dihydrochloride (DAPI) (#D9542, Sigma Aldrich, 1:1000 dilution). Secondary donkey anti-rabbit IgG antibodies either conjugated with an Alexafluor 488 (#A21206, Invitrogen) or 568 tag (#A10042, Invitrogen) were used for this study. The sections were imaged with a Nikon A1RS confocal microscope at the University of Wisconsin-Madison Optical Imaging Core. For each sample in this study, we collected seven images and analyzed these images using the 'Analyze Particles' application in NIH's ImageJ program as detailed in Darwisch et al.[36].

**Fibroblast cultures**. Primary fibroblasts were isolated from one-month-old mouse ears by our published protocol[13]. To briefly summarize the protocol, ears were collected from the mice into 1.5 ml microcentrifuge tubes containing 70% ethanol and rinsed with PBS containing penicillin and streptomycin antibiotics (Thermo Fisher Scientific, Waltham, MA). The tissues were cut into smaller pieces, and placed in a new tube with 0.5 ml Trypsin-EDTA (0.25% Trypsin, 0.1% EDTA) (Thermo Fisher Scientific, Waltham, MA) and 0.5 ml Dispase (5 U/ml) (STEMCELL Technologies, Vancouver, Canada), and incubated at 37 °C for 30 min. Preparations were centrifuged, and the supernatant was discarded. Tissues were washed with Hank's Balanced Salt Solution (Gibco) and centrifuged again, where the supernatant was discarded. Trypsin-EDTA was added to the tissue and allowed to incubate at 37 °C for 20 min. Following incubation, the samples were centrifuged, and the supernatant was removed from them. The remaining pellet was resuspended in Dulbecco's Modified Eagle's Medium (DMEM, ATCC, Manassas, VA) with 10% Fetal Bovine Serum (FBS, ATCC, Manassas, VA) and 1% Penicillin Streptomycin (Thermo Fisher Scientific, Waltham, MA). The pellet was triturated and then plated into a tissue culture dish and placed in a cell culture hood set at 37 °C with 5% CO₂. Fibroblasts were tested for mycoplasma contamination and the results were negative. Cultures were prepared for lysates for Western blot analysis or immunohistochemistry for PMP70. Lysates were made from the same number of cells using RIPA buffer containing protease inhibitors and quantified using a BCA Protein Assay Kit. For PEX14 and PMP70 immunohistochemistry, fibroblasts were allowed to grow on collagen-coated coverslips and then fixed with 4% PFA. The same western blot and immunohistochemistry protocol described above was used.

**Plasma parameter measurements**. Blood samples were collected from non-fasted mice by a submandibular bleed in EDTA capillary tubes and spun for 10 minutes at 1200 × g for 10 minutes. Plasma fraction was collected from the tube, placed into a separate microcentrifuge tube, and stored at −80 °C. Plasma cholesterol, triglycerides, and glucose concentrations were assayed and calculated with a Total Cholesterol Assay Kit (#STA-384, Cell Biolabs Inc, San Diego, CA), Triglyceride Assay Kit (#10010303, Cayman Chemical, Ann Arbor, MI), and Glucose Assay Kit (#10009582, Cayman Chemical, Ann Arbor, MI). Plasma ALT activity was quantified using an ALT Activity Assay Kit (#700620, Cayman Chemical, Ann Arbor, MI). We followed the protocols provided by the manufacturer.

**Liver histology**. 3-month-old mice were perfused using a 2% glutaraldehyde and 2% paraformaldehyde mixture before harvesting the largest liver lobe for paraffin sectioning. For livers collected for oil red o (ORO) staining, mice were perfused using a 4% paraformaldehyde fixative prior to subjecting the liver to a sucrose gradient and cryopreservation in Tissue-Tek O.C.T. Compound. The lobe was incubated in its respective fixative overnight and rinsed with PBS the following day. Samples were submitted to the University of Wisconsin–Madison's Translational Research Initiatives in Pathology (TRIP) core for processing and sectioning. Paraffin liver sections were stained with hematoxylin and eosin (H&E) and liver cryosections were stained with ORO using standard protocols.

**NAFLD phenotype scoring**. Five random different areas of each H&E-stained liver section were imaged using the tiling function of the Axio Imager 2 microscope (Carl Zeiss Microscopy LLC, White Plains, NY). In total, there were 125 images taken from each sample for scoring. We modified a scoring strategy based on the criteria for non-alcoholic fatty liver disease from the Pathology Committee of the NASH Clinical Research Network[81]. In this study, we focused on the presence of and total area affected by microvesicles, macrovacuoles, and hypertrophy. We scored a hepatic section for microvesicles as 1, macrovacuoles as 1, and hypertrophy as 1 if the pathology was present. We also scored if the pathologies encompassed 25–50% of the image as 1, 51–75% of the image as 2, and 76–100% of the image as 3. We summed the scores for each image and calculated the average of the image scores to generate the NALFD score for each mouse in this study. Two independent scorers evaluated liver images in this study.

**Metabolic cage phenotyping**. Male mice were acclimated to individual housing within metabolic phenotyping cages (Promethion Core, Sable Systems) with bedding and conditions identical to their home-cage environment. Stable body weight was confirmed 1 week prior to the metabolic phenotyping. For the study, mice were provided ad lib chow (#2016, Envigo, Madison, WI) and water. The environment was maintained at a constant temperature of 23 °C with a 12-hour light cycle that started at 7 A.M for 2.5 days. Oxygen, food, and water consumption as well as carbon dioxide production were measured continuously for the duration of the experiment. Energy expenditure was calculated using the Weir equation[82]. Body weight was measured at the start and end of the feeding trial. Body composition was measured using a Minispec LF-90 NMR machine (Bruker). Metabolic data were processed using the OneClick Macro (Sable Systems) to exclude outliers and non-physiological values. Data from only the final 48 hours (i.e. starting at 7 AM the morning after handling the mice) were used to avoid artifacts of animal handling. Hourly and daily average data were generated using CalR[83]. T tests and linear regression were performed using Prism 9.4 (GraphPad) to determine significant differences between the genotypes.

**Statistics and reproducibility**. One-way analysis of variance was performed in experiments using more than two groups and post hoc Tukey's honest significance test was utilized to determine which groups were significantly different from each other. A two-way Student's t test was used when there were only two experimental groups. All sample sizes are included within the figures and their legends. We used the G*Power application (RRID:SCR_013726) to determine sample sizes that would give us statistical power for 95% confidence with 80% power with an average standard deviation of 25%. We assumed all groups followed a normal distribution and had equal standard deviations. All statistical tests were performed using Prism Software (GraphPad, San Diego, CA). All statistical parameters can be found within the Supplementary Data file. We ensure the reproducibility of our results by obtaining similar conclusions from at least two independent experiments in this study.

**Reporting summary**. Further information on research design is available in the Nature Portfolio Reporting Summary linked to this article.

## Data availability

All lipidomic source data can be obtained through the Dryad public database under the accession code, doi:10.5061/dryad.vx0k6djvm. The raw mass spectrometry data will be given to those who are interested upon request to the corresponding author. Source data for figures and Supplementary Table 3 can be found in the Supplementary Data file.

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

## Acknowledgements

The authors would like to thank Satoshi Kinoshita and the University of Wisconsin (UW) Translational Research Initiatives in Pathology laboratory (TRIP), supported by the UW Department of Pathology and Laboratory Medicine, UWCCC (P30 CA014520) and the Office of The Director- NIH (S10OD023526) for the use of facilities and services, as well as Randall Massey and the University of Wisconsin Electron Microscope Core for tissue processing, sectioning, and assistance for this study. Confocal microscopy was performed at the University of Wisconsin-Madison Biochemistry Optical Core, which was established with support from the University of Wisconsin-Madison Department of Biochemistry Endowment. The authors want to recognize the laboratories of Dr. Freya Mowat and Janis Eells for their advice and feedback on this work. The authors would also like to extend their gratitude to Dr. Gregory Barrett-Wilt, Timothy Shriver, and the UW Biotechnology Center's Advanced Lipidomics Platform for their time and efforts in optimizing the protocols for our lipidomics experiments. The lipidomics work was supported in part by the UW Comprehensive Diabetes Center Core Services Pilot Award UWCDC-CSPA-20-7 and the UW Office of the Vice Chancellor for Research Graduate Education with funding from the Wisconsin Alumni Research Foundation. This work was also supported by grants from the National Eye Institute (R01EY022086 to A. Ikeda; P30EY016665 to the Department of Ophthalmology and Visual Sciences at the University of Wisconsin-Madison; NIH T32EY027721 to M. Landowski; F32EY032766 to M. Landowski; R01EY030513 to M-P Agbaga), Timothy William Trout Chairmanship (A. Ikeda), Research to Prevent Blindness Unrestricted grant to Dean McGee Eye Institute (M-P. Agbaga), and NIH grants S10OD028739, R01DK131742, and R01DK124696 (C.L.E.Yen).

## Author contributions

Conceptualization—M.L., S.I., and A.I. Data curation—M.L., V.J.B., S.G., Z.H., Y.K.G., M.T., R.S.B., L.J.M., D.W.N., C.R.D., and S.I. Formal analysis—M.L., V.J.B., Z.H., Y.K.G., M.T., R.S.B., L.J.M., D.W.N., and C.R.D. Funding acquisition—M.L., C.L.E.Y., M.P.A., and A.I. Investigation—M.L., C.L.E.Y., S.I., M.P.A., and A.I. Methodology—M.L., V.J.B., S.G., M.T., R.S.B., D.W.N., C.R.D., C.L.E.Y., S.I., M.P.A., and A.I. Project administration—C.L.E.Y., S.I., M.P.A., and A.I. Resources—M.L., M.T., R.S.B., D.W.N., C.L.E.Y., S.I., M.P.A., and A.I. Supervision—A.I. Validation—M.L., V.J.B., S.G., Z.H., Y.K.G., M.T., R.S.B., L.J.M., D.W.N., C.R.D., S.I., M.P.A., and A.I. Visualization—M.L., V.J.B., S.G., D.W.N., S.I., and A.I. Writing—original draft—M.L., S.I., and A.I. Writing—review and editing—all authors have contributed to the review and editing of this manuscript.

## Competing interests

The authors declare no competing interests.
