## [Peer Review File · Communications Biology]

Reviewers' comments:

Reviewer #1 (Remarks to the Author):

The authors report interesting observations pertaining to the role of TMEM135, which is a protein that has been located in peroxisomes, mitochondria and in lipid droplets. The findings are novel and intriguing and reveal that TMEM135 plays an essential role in lipid homeostasis, in particular of PUFA. The hypothesis to explain the findings, i.e. that TMEM135 transports DHA and DPA out of the peroxisome after synthesis, is attractive and if true, it would constitute a missing link in PUFA metabolism. This will need to be substantiated in future research.

The abstract is rather confusing because the connection between the beneficial effect of defective TMEM135 in leptin mice and the PUFA dysregulation in TMEM135 knockout mice was not made clear. Also in the rest of the manuscript the links between the different findings should be better explained and emphasized.

Throughout the text it is claimed that loss of TMEM135 increases peroxisomal β -oxidation. This is however only based on increased expression of one peroxisomal β -oxidation enzyme, namely Acox1. The authors also mention increased expression of catalase, but this H₂O₂ degrading enzyme does not belong to the peroxisomal β -oxidation pathway. Its upregulation is rather an argument for general proliferation of peroxisomes. It is thus an overstatement that peroxisomal β -oxidation is increased. More evidence needs to be provided.

In the discussion the authors consider that the improved lipid homeostasis in ob/ob mice, in which TMEM135 was deleted, could be mediated by activation of PPAR α . This is something that can easily be assessed by checking the transcripts of a number of PPAR α target genes, encoding not only peroxisomal but also ER and mitochondrial proteins. This should be done.

The way peroxisome abundance was assessed needs to be optimized. First, the golden standard to detect peroxisomes at the ultrastructural level is to do the alkaline DAB staining to localize catalase activity. Without this, the structures seen in the micrographs are unsure to represent peroxisomes. Secondly, PMP70 is used as a marker for peroxisome abundance. In fact, this transporter is regulated quite strongly, also by PPAR α (some references are PMID 18585430, 17978498, 17686565), and performing immunofluorescence or immunoblotting for this protein is not the best option. It is rather recommended to use PEX14 (which is shown in supplementary data) as a protein that is closely linked with peroxisome numbers.

Minor remarks

Line 75: "Peroxisomes can also generate ether lipids for plasmalogen synthesis" – plasmalogens ARE etherlipids, please specify better what plasmalogens are

To me it is not very informative how many lipid species in a certain class are changed (pie diagrams). It is more important which FA they contain and whether the class is changed as a whole.

Line 283 -285: an explanation is given why a single peroxisomal β -oxidation step is needed to produce DPA and DHA. It is however not because the ER lacks the enzymes; it should be rather clarified why first a C24 fatty acid is made from C18 precursors, which is subsequently shortened. "In contrast, the levels of ether lipids correlated with peroxisome number" this is not accurately formulated. Which ether lipids and what is the correlation?

Reviewer #2 (Remarks to the Author):

TMEM135 in Peroxisomes review:

"Peroxisomes are tasked with critical lipid metabolic roles within eukaryotic cells". In this article, the authors study the role of transmembrane protein 135 (TMEM135) on peroxisome function and lipid metabolism. The use of the ob/ob mouse to study TMEM135 function in peroxisomes is quite interesting, as the phenotype of the ob/ob mouse has been previously well characterized with obesity, dyslipidemia, and fatty liver.

Critical findings in this paper include the fact that the ob/ob phenotype is ameliorated with TMEM135 knockdown. The authors provide histological evidence, and functional evidence of improvement in the NAFLD phenotype. Further, TMEM135 seems critical for peroxisomal PUFA

metabolism and peroxisomal biogenesis. The findings are novel, they appear to be true. There is a potential clinical relevance to several human conditions involving disturbances in cellular lipid homeostasis.

Few minor concerns remain:

The claims in this manuscript are worth publishing. Although the manuscript is straight forward and well written, a few concerns remain. From the data provided, TMEM135 seems to affect peroxisomal enzyme levels, and DHA/DPA levels in a manner that does not correlate with peroxisomal number. In addition, non-peroxisomal mechanisms have not been explored. However, they could be partly contributing to the liver phenotype.

The manuscript, at least, could be strengthened by the following:

1. Seahorse studies/ Mitochondrial respiration in liver cells
2. Assessment of oxidative stress response
3. Assessment of ER stress signaling in the different double-mutants.
4. TMEM135 is highly expressed in brain tissue. Additional studies are needed to assess caloric intake and energy expenditure, particularly in the setting of improved weight gain.
5. Validation of the biological findings in vitro using liver cells in culture, particularly in the context of TMEM135 levels and peroxisomal biogenesis.

The above could also increase our understanding of a mechanism for the liver phenotype. A statement or paragraph on the limitations potentially associated with this type of research would be useful.

Other minor points:

Both the TMEM135 overexpressing TG mice and TMEM135 knockdown were used in the study. Throughout the manuscript, the expression "TMEM135 mutant" is often used. The reader could benefit if the manuscript specified when *Tmem135*^{FUN025/FUN025} or the TMEM135 TG is used.

Dear Editor and Reviewers,

We thank you for your feedback and comments on our manuscript. Please find our responses to your comments below and revisions to our manuscript in red font.

Response to Reviewer #1's comments:

The authors report interesting observations pertaining to the role of TMEM135, which is a protein that has been located in peroxisomes, mitochondria and in lipid droplets. The findings are novel and intriguing and reveal that TMEM135 plays an essential role in lipid homeostasis, in particular of PUFA. The hypothesis to explain the findings, i.e. that TMEM135 transports DHA and DPA out of the peroxisome after synthesis, is attractive and if true, it would constitute a missing link in PUFA metabolism. This will need to be substantiated in future research.

Response: We appreciate the kind words of Reviewer 1 about our intriguing observations described in the manuscript. In our revised manuscript, we have added more data to strengthen the finding that TMEM135 functions in peroxisomal PUFA, specifically DHA, metabolism. This is the summary of the data we have added to our revised manuscript:

1) Lipidomics analysis of the plasmas from 2.5-month-old wild-type (WT), *Tmem135* TG, and *Tmem135*^{FUN025/FUN025} mice. We found a similar decrease of DHA-containing lipids in *Tmem135*^{FUN025/FUN025} plasmas as we previously observed in the livers, retinas, and hearts of *Tmem135*^{FUN025/FUN025} mice compared to WT. (see Figure 1 and S1 as well as Tables S1, S2, and S9)

2) Fatty acid composition analysis of the livers, retinas, hearts, and plasmas from 2.5-month-old WT and *Tmem135*^{FUN025/FUN025} mice. We detected significant decreases in DHA concentrations in the livers, retinas, hearts, and plasmas of *Tmem135*^{FUN025/FUN025} mice. (see Figure 2 and Table S3)

3) Analysis of the “Sprecher pathway” of DHA synthesis in *Tmem135*^{FUN025/FUN025} mice. The only changes of Sprecher pathway components detected in our study was significant increases of peroxisomal beta-oxidation enzymes ACOX1, DBP, and ACAA1. These three proteins are required for the generation of DHA in peroxisomes. Our results show increases of peroxisomal beta-oxidation enzymes. (see Figure 3) along with the DHA decreases in the *Tmem135*^{FUN025/FUN025} liver.

The abstract is rather confusing because the connection between the beneficial effect of defective TMEM135 in leptin mice and the PUFA dysregulation in TMEM135 knockout mice was not made clear. Also in the rest of the manuscript the links between the different findings should be better explained and emphasized.

Response: We hypothesize that the protection provided by the *Tmem135* mutation on leptin mutant mouse phenotypes is due to increased peroxisomal oxidation induced by altered DHA metabolism. We have added new data that supports this claim. This data includes:

1) Lipidomics analysis of the plasmas from 3-month-old *Lep*^{ob/ob} and *Tmem135*^{FUN025/FUN025}/*Lep*^{ob/ob} mice. We observed significant decreases of DHA-containing lipids in the *Tmem135*^{FUN025/FUN025}/*Lep*^{ob/ob} plasmas relative to *Lep*^{ob/ob} plasmas. (see Figure 8 and S7 as well as Tables S4 and S5)

2) Western blot analysis of peroxisomal beta-oxidation enzymes in 3-month-old male and female *Lep*^{ob/ob}, TG/*Lep*^{ob/ob}, and *Tmem135*^{FUN025/FUN025}/*Lep*^{ob/ob} livers. We measured higher levels of ACOX1, DBP, and ACAA1 in male and female *Tmem135*^{FUN025/FUN025}/*Lep*^{ob/ob} livers. (see Figure 8 and Figure S8).

We have also extensively edited our manuscript including our abstract, results, and discussion sections to better explain different findings in our study. One major change we made to our revised manuscript was the reversal of the figure order in our results section. The results section in our revised manuscript begins with our lipidomics analysis and ends with the findings from our *Lep^{ob/ob}* mouse studies. We believe the reverse order of our figures has greatly improved the flow and transitions between the findings we present in our manuscript. These changes can be reflected in our revised abstract that now reads:

*Dysregulation of lipid metabolism is a hallmark of multiple human diseases. Here, we study the role of transmembrane protein 135 (TMEM135) in lipid metabolism. While TMEM135 is hypothesized to act in the cellular response to increased intracellular lipids, no defined molecular function for TMEM135 has been identified. We performed a lipidomics and fatty acid analysis of tissues from *Tmem135* mutant mice and found striking reductions in docosahexaenoic acid (DHA) across all *Tmem135* mutant tissues. Further investigation also revealed increases in the peroxisomal number and peroxisomal beta-oxidation enzymes due to the *Tmem135* mutation. The significance of these *Tmem135* mutation-induced cellular changes in metabolic disease was evaluated using leptin mutant mice in which TMEM135 was found to be upregulated. We found that a mutation in the *Tmem135* gene ameliorates the metabolic disease phenotypes of leptin mutant mice including obesity, dyslipidemia, and non-alcoholic fatty liver. This protection may be explained by increased peroxisomal beta-oxidation due to the *Tmem135* mutation. Thus, we conclude that TMEM135 has a novel critical function in lipid homeostasis through its role in peroxisomal DHA metabolism and mediates the development of metabolic disease phenotypes.*

We have also better explained our findings in the discussion of our manuscript and specifically added more details about the protection of the *Tmem135* mutation on leptin mutant mouse phenotypes. We have also made edits to our **Figure 9** and its figure legend which is a schematic of the proposed TMEM135 function in peroxisomes.

Throughout the text it is claimed that loss of TMEM135 increases peroxisomal β -oxidation. This is however only based on increased expression of one peroxisomal β -oxidation enzyme, namely Acox1. The authors also mention increased expression of catalase, but this H₂O₂ degrading enzyme does not belong to the peroxisomal β -oxidation pathway. Its upregulation is rather an argument for general proliferation of peroxisomes. It is thus an overstatement that peroxisomal β -oxidation is increased. More evidence needs to be provided.

Response: We have surveyed the levels of three additional peroxisomal β -oxidation proteins (DBP, ACAA1, and SCPx) using Western blot analysis of the liver lysates used in this study. This data is included in **Figure 3, 8, and S8**. In short, we found increased DBP and ACAA1 protein but no changes in SCPx protein in both *Tmem135^{FUN025/FUN025}* and *Tmem135^{FUN025/FUN025}/Leptin^{ob/ob}* livers compared to controls. Since ACOX1, DBP, ACAA1, and SCPx are required for peroxisomal beta-oxidation (PMID: 26858947), our results support increased hepatic peroxisomal beta-oxidation occurs due to the *Tmem135* mutation in mice. We believe this is ample evidence that supports our claim that the *Tmem135* mutation increases peroxisomal beta-oxidation.

We also agree with Reviewer 1's comment about catalase and have regarded catalase as a marker for general peroxisome proliferation in both *Tmem135^{FUN025/FUN025}* and *Tmem135^{FUN025/FUN025}/Leptin^{ob/ob}* livers in this study. This data can be found in **Figures S2, 8, and S8**.

In the discussion the authors consider that the improved lipid homeostasis in ob/ob mice, in which TMEM135 was deleted, could be mediated by activation of PPAR α . This is something that can easily be assessed by checking the transcripts of a number of PPAR α target genes, encoding not only peroxisomal but also ER and mitochondrial proteins. This should be done.

Response: In our revised manuscript, we reported increases in ACOX1, DBP, and ACAA1 (**Figure 3d, 8c, and S8a**), all of which are downstream targets of PPAR α signaling (PMID: 20936127). We also reported decreases in CPT1a (**Figure 5 and S9**), which is also a downstream target of PPAR α signaling (PMID: 20936127). These findings are inconclusive on whether PPAR α signaling is or is not active in both *Tmem135*^{FUN025/FUN025} and *Tmem135*^{FUN025/FUN025}/*Leptin*^{ob/ob} livers. To examine a role of PPAR α signaling in the *Tmem135*^{FUN025/FUN025} livers, we crossed these mice with *Ppara* knockout mice to generate *Tmem135*^{FUN025/FUN025}/*Ppara*^{-/-} mice. We evaluated the levels of PEX14 and PMP70 by Western blot analysis using livers from 3.0-month-old *Tmem135*^{FUN025/FUN025} and *Tmem135*^{FUN025/FUN025}/*Ppara*^{-/-} mice. We found PEX14 and PMP70 were decreased in the *Tmem135*^{FUN025/FUN025}/*Ppara*^{-/-} livers compared to *Tmem135*^{FUN025/FUN025} livers (**Figure 4e**). We confirm these results by showing decreased PEX14 immunoreactivity on *Tmem135*^{FUN025/FUN025}/*Ppara*^{-/-} liver cryosections compared to *Tmem135*^{FUN025/FUN025} liver cryosections (**Figure 4f**). These results show PPAR α signaling contributes to peroxisome concentrations in *Tmem135*^{FUN025/FUN025} livers. Thus, it is possible that the protection of the *Tmem135* mutation on leptin mutant mouse phenotypes could be mediated through PPAR α signaling.

The way peroxisome abundance was assessed needs to be optimized. First, the golden standard to detect peroxisomes at the ultrastructural level is to do the alkaline DAB staining to localize catalase activity. Without this, the structures seen in the micrographs are unsure to represent peroxisomes. Secondly, PMP70 is used as a marker for peroxisome abundance. In fact, this transporter is regulated quite strongly, also by PPAR α (some references are PMID 18585430, 17978498, 17686565), and performing immunofluorescence or immunoblotting for this protein is not the best option. It is rather recommended to use PEX14 (which is shown in supplementary data) as a protein that is closely linked with peroxisome numbers.

Response: We agree with Reviewer 1 and optimized the way to assess peroxisome abundance. We understand that the gold standard identification of peroxisomes on electron micrographs is by alkaline DAB staining. Unfortunately, we were unable to find assistance at the University of Wisconsin-Madison to help us with this staining procedure as it is a technique unfamiliar to us. Therefore, we have removed these electron micrograph images from our revised manuscript. Instead, we have decided to use PEX14 as a quantitative marker that is closely linked with peroxisome number as recommended by the reviewer to assess peroxisome abundance. We have revised our **Figure 4** depicting hepatic peroxisome proliferation to include data on both PEX14 and PMP70. Both of these common peroxisome markers are increased in *Tmem135*^{FUN025/FUN025} livers. Furthermore, **Figure 8d and S8b** show similar results of increased PEX14 and PMP70 in the livers of male and female *Tmem135*^{FUN025/FUN025}/*Leptin*^{ob/ob} mice. Thus, our newly revised **Figures 4, 8d, and S8b** convincingly display data showing loss of TMEM135 increases peroxisome proliferation in the mouse liver. All other figures including data only on PMP70 have been moved from the main figures and are now included in supplemental figures (i.e. **Figure S3 and S4**).

Minor remarks

Line 75: “Peroxisomes can also generate ether lipids for plasmalogen synthesis” – plasmalogens ARE etherlipids, please specify better what plasmalogens are

Response: We apologize to Reviewer 1 about our textual mistake and have corrected it in our revised manuscript:

Peroxisomes can also generate 1-O-alkyl glycerol-3-phosphates for plasmalogen synthesis¹ and docosahexaenoic acid (DHA) from essential dietary fatty acids² while the endoplasmic reticulum (ER) produces membrane lipids³.

To me it is not very informative how many lipid species in a certain class are changed (pie diagrams). It is more important which FA they contain and whether the class is changed as a whole.

Response: We agree with Reviewer 1 that the pie graphs included in our previous manuscript were not very informative. We have included two supplemental tables in our revised manuscript. In **Table S1**, we have summarized our quantitation of the total number of lipids that were detected, significantly upregulated, and significantly downregulated per lipid class in *Tmem135*^{FUN025/FUN025} tissues compared to WT. In **Table S2**, we performed a similar analysis where we determined the total number of lipids that were detected, significantly increased, and significantly decreased per fatty acid side group in *Tmem135*^{FUN025/FUN025} tissues compared to WT. We also did this for our new lipidomics data from *Lep*^{ob/ob} and *Tmem135*^{FUN025/FUN025}/*Lep*^{ob/ob} plasmas which can be found in **Table S4** and **S5**.

In our revised manuscript, we also include data on the fatty acid concentrations in the livers, retinas, hearts, and plasmas of 2.5-month-old WT and *Tmem135*^{FUN025/FUN025} tissues. This data can be found in **Figure 2** and **Table S3**.

Line 283 -285: an explanation is given why a single peroxisomal β -oxidation step is needed to produce DPA and DHA. It is however not because the ER lacks the enzymes; it should be rather clarified why first a C24 fatty acid is made from C18 precursors, which is subsequently shortened.

Response: To better explain the process of DHA synthesis, we have included a discussion of the “Sprecher pathway” of DHA synthesis in the results and discussion sections of our revised manuscript. This pathway encompasses a series of elongation and desaturation steps at the ER to generate C24:6n3. Then, C24:6n3 is transferred to peroxisomes and retroconverted to C22:6n3 through one round of its beta-oxidation machinery. These processes in the ER and peroxisome are needed to complete the synthesis of DHA (PMID: 10471115). We also have a schematic of the Sprecher pathway included with our **Figure 3** where we investigate the levels of the components of the Sprecher pathway.

“In contrast, the levels of ether lipids correlated with peroxisome number” this is not accurately formulated. Which ether lipids and what is the correlation?

Response: We have included this information about ether lipids when we refer to PPAR activating ligands. This section reads:

Known ligands for PPARs, ether phosphatidylethanolamine (EtherPEs)¹⁰, can be produced by peroxisomal beta-oxidation (Fig. 9)⁵⁶, and are increased in *Tmem135* mutant tissues (**Table S1**). In fact, EtherPE16:1e_18:1 and 16:1e_18:2 are commonly increased across *Tmem135* mutant tissues (**Tables S6-9**). These data suggest the possibility that increased EtherPEs may activate PPAR signaling in *Tmem135* mutant mice.

Response to Reviewer #2's comments:

TMEM135 in Peroxisomes review:

"Peroxisomes are tasked with critical lipid metabolic roles within eukaryotic cells". In this article, the authors study the role of transmembrane protein 135 (TMEM135) on peroxisome function and lipid metabolism. The use of the ob/ob mouse to study TMEM135 function in peroxisomes is quite interesting, as the phenotype of the ob/ob mouse has been previously well characterized with obesity, dyslipidemia, and fatty liver.

Critical findings in this paper include the fact that the ob/ob phenotype is ameliorated with TMEM135 knockdown. The authors provide histological evidence, and functional evidence of improvement in the NAFLD phenotype.

Further, TMEM135 seems critical for peroxisomal PUFA metabolism and peroxisomal biogenesis. The findings are novel, they appear to be true. There is a potential clinical relevance to several

human conditions involving disturbances in cellular lipid homeostasis.

Response: We agree with Reviewer 2 about the significance of our finding that there was amelioration of the *Lep^{ob/ob}* mouse phenotype due to the *Tmem135* mutation. We have added new data on the levels of plasma apolipoproteins APOB and APOA1 that shows the *Tmem135* mutation reduces plasma APOB100 (**Figure 7**), a marker of atherogenic lipoproteins in humans (PMID: 34677405), in leptin mutant mice. This adds to the potential clinical relevance of our study to several metabolic diseases in humans.

Few minor concerns remain:

The claims in this manuscript are worth publishing. Although the manuscript is straight forward and well written, a few concerns remain. From the data provided, TMEM135 seems to affect peroxisomal enzyme levels, and DHA/DPA levels in a manner that does not correlate with peroxisomal number.

Response: Our data shows that the *Tmem135* mutation affects peroxisomal enzyme levels (increased) in a manner that correlates with peroxisome number (increased). We have confirmed the increase in peroxisome number by including our data of PEX14 in **Figure 4** of our revised manuscript. In contrast, DHA levels are affected in a manner that does not correlate with peroxisomal number. Despite the increased number of peroxisomes, which play critical roles in DHA synthesis, the DHA levels are significantly decreased in *Tmem135* mutant mice. In fact, these contrasting findings prompted us to hypothesize that TMEM135 has a role in the export of DHA from peroxisomes following synthesis, which is impaired by the *Tmem135* mutation leading to the decrease in DHA.

In addition, non-peroxisomal mechanisms have not been explored. However, they could be partly contributing to the liver phenotype.

Response: We agree with Reviewer 2 about the possibility that non-peroxisomal mechanisms may be contributing to the protection of the *Tmem135* mutation on the leptin mutant mouse phenotypes. In our previous manuscript, we found decreases in CPT1a and increases in CACT in both *Tmem135^{FUN025/FUN025}* and *Tmem135^{FUN025/FUN025}/Leptin^{ob/ob}* livers. However, we realize we did not explain and present our results well. We have included our Western blot analysis of mitochondrial proteins in **Figure 5** and **S9** of our revised manuscript where the changes in CPT1a and CACT protein are more apparent to readers. We have also better explained these results in our discussion section. This portion of the discussion from our revised manuscript reads:

The impact of the Tmem135 mutation on other organelles that are known to harbor TMEM135 such as mitochondria^{17,20} and lipid droplets^{20,68} could possibly explain the decreased severity of leptin mutant mouse phenotypes. We detected changes in mitochondrial proteins CPT1a and CACT but no differences in Perilipin 2 (PLIN2) (Fig. S10), the main protein constituent of hepatic lipid droplets⁶⁹, in the livers of mice with a Tmem135 mutation. The origin of mitochondrial changes may be traced back to peroxisomes. For example, the decreased CPT1a protein due to the Tmem135 mutation may be a compensatory change to lower mitochondrial fatty acid oxidation since there is higher peroxisomal fatty acid oxidation enzymes in these livers. CPT1a is the rate-limiting step for mitochondrial fatty acid oxidation that converts acyl-CoA esters to acylcarnitines for their import through the mitochondrial outer membrane⁴¹. It has been shown that liver-specific Cpt1a knockout mice were protected against diet-induced weight gain⁷⁰. Thus, decreased hepatic CPT1a concentrations resulting from the Tmem135 mutation may explain the smaller body weight of Tmem135 and leptin double mutant mice compared to leptin mutant controls. The findings from this study support our previous work that shows modifications in Tmem135 function have profound effects on mitochondrial homeostasis^{17-19,21,22}. Future studies are warranted to investigate whether the mitochondrial dysfunction caused by the Tmem135 mutation is from its role on peroxisomes or

mitochondria as this may shed insight into the mechanisms underlying its protection against the leptin mutant mouse phenotypes.

The manuscript, at least, could be strengthened by the following:

Response: We recognize Reviewer 2's suggestion for us to perform additional experiments to strengthen our manuscript. We believe the new data added and textual edits made to our revised manuscript have strengthened our conclusions that the *Tmem135* mutation protects against *Lep^{ob/ob}* mouse phenotypes. We believe these changes allow us to make a compelling case that the benefit of the *Tmem135* mutation in leptin mutant mice is due to decreasing DHA incorporation into lipids and increasing DHA oxidation within peroxisomes. We refer Reviewer 2 to our **Figure 8** and **S8** that includes this new data.

We appreciate Reviewer 2's suggestions for experiments to further add to the mechanism explaining the protection of the *Tmem135* mutation on leptin mutant liver phenotypes. We have completed some of these experiments and present our findings below:

1. Seahorse studies/ Mitochondrial respiration in liver cells

Response: We have not completed these experiments because the preparation and culturing of primary hepatocytes is technically challenging as described in PMID: 33111119. We recognize previous studies have isolated and cultured hepatocytes from leptin mutant mice (PMID: 32937194 and 27432632). However, if we were to observe mitochondrial respiration changes using a Seahorse Xfe24 extracellular flux analyzer, we would be unable to discern between the contributions of peroxisomes and mitochondria on these mitochondrial respiration changes, since the Seahorse assay can detect the contributions of both peroxisomes (PMID: 35733440) and mitochondria (PMID: 28276021).

2. Assessment of oxidative stress response

Response: We evaluated the levels of two oxidative stress signaling proteins, (a) Superoxide dismutase 2 (SOD2) and (b) Glutathione peroxidase 2 (GPX2), in the 3.0-month-old male *Lep^{ob/ob}* (*ob*), *Tmem135* TG/*Lep^{ob/ob}* (TG/*ob*), and *Tmem135^{FUN025/FUN025}*/*Lep^{ob/ob}* (*FUN025/ob*) livers. However, we did not see any changes in SOD2 and GPX2 between the *Lep^{ob/ob}* and *Tmem135^{FUN025/FUN025}*/*Lep^{ob/ob}* mice that could explain the protection of the *Tmem135* mutation on *Lep^{ob/ob}* phenotypes.

Western blot analysis of (a) Superoxide dismutase 2 (SOD2) and (b) Glutathione peroxidase 2 (GPX2) using livers from 3-month-old male *Lep^{ob/ob}* (N=3), *Tmem135* TG/*Lep^{ob/ob}* (N=3) and *Tmem135^{FUN025/FUN025}*/*Lep^{ob/ob}* (N=3) mice. Data is presented mean ± SD. Dots represent individual data points. * indicates post hoc Tukey test for a P<0.05 significance following a significant difference detected by one-way ANOVA. Dots represent individual data points.

3. Assessment of ER stress signaling in the different double-mutants.

Response: We also evaluated the levels of Protein disulfide-isomerase (PDI) in the 3.0-month-old male *Lep^{ob/ob}* (*ob*), *Tmem135* TG/*Lep^{ob/ob}* (TG/*ob*), and *Tmem135^{FUN025/FUN025}/Lep^{ob/ob}* (*FUN025/ob*) livers. However, we did not see any changes in PDI between the *Lep^{ob/ob}* and *Tmem135^{FUN025/FUN025}/Lep^{ob/ob}* mice.

Western blot analysis of Protein disulfide-isomerase (PDI) using livers from 3-month-old male *Lep^{ob/ob}* (N=3), *Tmem135* TG/*Lep^{ob/ob}* (N=3) and *Tmem135^{FUN025/FUN025}/Lep^{ob/ob}* (N=3) mice. Data is presented mean ± SD. Dots represent individual data points. * indicates post hoc Tukey test for a P<0.05 significance following a significant difference detected by one-way ANOVA. Dots represent individual data points.

4. TMEM135 is highly expressed in brain tissue. Additional studies are needed to assess caloric intake and energy expenditure, particularly in the setting of improved weight gain.

Response: We agree with Reviewer 2 that *Tmem135* expression is high in the brains of mice as we found in a previous study (PMID: 33064130). We also performed a fatty acid analysis of the brains from 2.5-month-old male WT and *Tmem135^{FUN025/FUN025}* mice and found this tissue also had a significant decrease in DHA concentrations (see below). We are currently investigating if there is a pathological or behavioral phenotype in *Tmem135* mutant mice that could accompany their decreased DHA levels. It would be interesting to see if the *Tmem135* mutation has a role in caloric intake and energy expenditure too. Ongoing studies are examining this research question and beyond the scope of this manuscript.

Bar graph of relative moles of docosahexaenoic acid (DHA, C22:6n3) quantified by gas chromatography mass spectrometry in four 2.5-month-old male WT and *Tmem135^{FUN025/FUN025}* (*FUN025*) brains. Data is presented mean ± SD. **** indicates P<0.0001 significance by two-way Student's T-test.

5. Validation of the biological findings in vitro using liver cells in culture, particularly in the context of TMEM135 levels and peroxisomal biogenesis.

Response: Instead of establishing primary hepatocyte cultures from WT, *Tmem135* TG, and *Tmem135^{FUN025/FUN025}* mice, we have cultured fibroblasts from these mice and examined their peroxisome amounts. We observed that the fibroblasts from *Tmem135* TG have lower peroxisome numbers whereas the fibroblasts from *Tmem135^{FUN025/FUN025}* mice have higher peroxisome numbers through Western blot analysis and immunohistochemistry for PMP70 (Figure S4). These findings indicate the changes in peroxisomes due to *Tmem135* function is from a cell autonomous effect and not from systemic influences.

Therefore, we validated our biological findings that TMEM135 levels mediate peroxisome numbers both *in vivo* and *in vitro*.

The above could also increase our understanding of a mechanism for the liver phenotype. A statement or paragraph on the limitations potentially associated with this type of research would be useful.

Response: We agree with Reviewer 2 that it is important to specify the limitations of this type of research. We have included a statement in our discussion about the need to perform additional experiments to identify the functional contribution of TMEM135 on peroxisomes and mitochondria in cellular homeostasis, especially regarding the protection of the *Tmem135* mutation against the leptin mutant mouse phenotypes.

*Future studies are warranted to investigate whether the mitochondrial dysfunction caused by the *Tmem135* mutation is from its role on peroxisomes or mitochondria as this may shed insight into the mechanisms underlying its protection against the leptin mutant mouse phenotypes. It is also possible that the protection of the *Tmem135* mutation on leptin mutant mouse phenotypes is from its effect on another tissue. We have shown in our study that multiple tissues in *Tmem135* mutant mice exhibit similar changes in DHA concentrations and peroxisomal proteins. Therefore, additional experiments are required to determine the contribution of tissues towards the protection of the *Tmem135* mutation on the phenotypes of leptin mutant mice.*

Other minor points:

Both the TMEM135 overexpressing TG mice and TMEM135 knockdown were used in the study. Throughout the manuscript, the expression "TMEM135 mutant" is often used. The reader could benefit if the manuscript specified when *Tmem135*^{FUN025/FUN025} or the TMEM135 TG is used.

Response: In this study, we use mice that overexpress wild-type *Tmem135* (*Tmem135* TG) and mice with a mutation in the donor splice site of exon 12 (*Tmem135* mutant or *Tmem135*^{FUN025/FUN025}). We clarified when we used *Tmem135* TG and *Tmem135*^{FUN025/FUN025} in our revised manuscript. We also have reduced our discussion of the *Tmem135* TG mice in our revised manuscript to avoid confusion for readers.

Reviewers' comments:

Reviewer #1 (Remarks to the Author):

The authors extensively worked on the manuscript and provided new data. Overall, the reversal of the results section is more logic. However, as already remarked before, the link between the first part (Figures 1 – 4) and the second part (Figures 6 -9) is not clear and it is more obvious now that these are two different stories. The first one, with the observation that DHA levels are strongly decreased in TMEM135 deficient mice despite increased expression of peroxisomal beta-oxidation enzymes is intriguing. Regrettably, only an assumption can be made on the underlying mechanism and thus this line of thinking is unfinished. In the second part, the positive effect of TMEM135 deletion on ob/ob mice and fatty liver disease is shown, but the causative role of increased peroxisomal beta-oxidation and of reduced DHA levels is not demonstrated. Because TMEM135 is also expressed in other cellular compartments (mitochondria, lipid droplets), other mechanisms can also play a role. The lack of conclusions that can be drawn from both parts of the manuscript is very evident in the abstract.

Abstract

Mutation in TMEM135 causes both a reduction in DHA and an increase in peroxisomal beta-oxidation enzymes: without any further explanation or hypothesis, this is contradictory according to the present knowledge and very confusing. In addition, it remains unclear whether the positive effect of the TMEM135 mutation in ob/ob mice is also mediated by a reduction in DHA

Introduction

Line 90-92 and sentence of subsequent paragraph should be reorganized

Results

Line 128: when analyzing acyl side chains of the lipids, only strong decreases in DHA are stated. Although it is OK to focus on this PUFA in the rest of the text, at least it should be mentioned what happens with the levels of other PUFA (including AA) and saturated fatty acids.

Figure 3d: the peroxisomal enzymes ACOX1, DBP and ACAA1 are processed into smaller fragments after their import in peroxisomes. In the figure of the western blot, it is not mentioned which is the size of the protein that is detected.

Figure 4: from the images and the western blot, it is impossible to deduce that there are more peroxisomes present per cell and thus to conclude that there is peroxisome proliferation. Much larger magnifications are needed in order to visualize individual peroxisomes such that they can be counted. ICC is usually better suited to check peroxisomal numbers than IHC.

Figure 5: CPT1A is reduced in TMEM135/FUN025/FUN025 livers. It is well known that CPT1A is a PPAR α target gene (PMID: 20638986, PMID: 29795111 and ref 59 of the manuscript). How is this reconciled with the claim that PPAR α is activated? What was the expression of CPT1A in the TMEM135/PPAR α knockouts? Because of the opposing changes in CPT1 α and CACT and other mitochondrial import proteins that are unaffected, this paragraph is not very informative. To further prove PPAR α activation in liver the ER enzyme CYP4A10, that is strongly regulated by PPAR α , would be a good option.

Line 235: cholesterol is reduced in double TMEM/ob/ob mice. How are these data explained in light of changes in peroxisomes?

Line 288: what are 'numbers' referring to?

Discussion:

Line 303 -306. Several mouse models are mentioned in which DHA is lacking. If peroxisomes are essential for DHA synthesis, are there no mouse models in which peroxisomal beta-oxidation is deficient?

Line 319-321: can the authors be more specific how the phenotypes of mouse models with beta-oxidation deficiency differ from TMEM135 deficient mice?

Line 383-385: the suggestion that mitochondrial beta-oxidation would be down regulated when peroxisomal beta-oxidation is upregulated, is not very likely. These two beta-oxidation pathways are not redundant and serve different functions. For the breakdown of the common fatty acids (C16 -C18), mitochondrial beta-oxidation is much more adapted and, in contrast to peroxisomal beta-oxidation, energy generating. As mentioned before, normally peroxisomal and mitochondrial beta-oxidation are coordinately upregulated by PPAR α (see also refer 59).

Minor comments

Line 150: 'determined'

Line 321: based 'off' these results

There are more textual flaws

Reviewer #2 (Remarks to the Author):

TMEM135 in Peroxisomes review:

This is a resubmission Pinel et al. On the first submission, we had several concerns and questions that have been reasonably addressed. One major concern remaining. Some of the data, as presented will elicit a variety of controversies in the field.

1. Omega-3 long chain PUFA supplementation has a beneficial effect on obesity, diabetes, cardiovascular disease, thermogenic function of adipocytes, and counteract the effects of omega-6 long chain PUFA. So, in the clinical realm, we try to supplement with Omega-3 to mitigate the ill effects of obesity and excess fat. The major concern here is, why in the setting of a TMEM135 mutation (presumed knockdown), reduced DHA is beneficial?

2. I again agree with publication of the data, although the DHA story is still counterintuitive. In any event, the data seem true, novel, and relevant to the clinical interest, in terms of a role for TMEM135 in lipid accumulation in the setting of a variety of illnesses.

3. As the authors know the role of DHA's in health and illnesses has emerged as a critical clinical piece. Nevertheless, the mechanisms of how Omega-3 long chain PUFAs help with reduced inflammation or help with obesity and heart disease are still unclear. So, altered DHA metabolism may be totally incidental, and not the true cause for TMEM135 knockdown help.

4. In essence, this is reasonable publishable piece of research, but the authors have not completely delineated a clear mechanism. There are several unanswered questions. The authors cannot address them all in one manuscript, but few basic questions should be answered. For instance, do the authors know if caloric expenditure and food intake are altered in these mice? Is this part of future studies?

5. Perhaps in the discussion section, the authors can better highlight the potential controversies and soften the tone, indicating that mitigating effects for the ob/ob_TMEM135 knockdown is perhaps NOT through alteration of DHA metabolism. Those non-DHA ideas will be further explored in subsequent manuscripts.

6. In summary, the paper can be accepted with some revision of the discussion section.

Dear Editor and reviewers,

We thank you for your feedback and comments on our revised manuscript. Please find our responses to the comments below and edits to our revised manuscript in red font.

Response to Reviewer #1's comments:

The authors extensively worked on the manuscript and provided new data. Overall, the reversal of the results section is more logic. However, as already remarked before, the link between the first part (Figures 1 – 4) and the second part (Figures 6 -9) is not clear and it is more obvious now that these are two different stories. The first one, with the observation that DHA levels are strongly decreased in TMEM135 deficient mice despite increased expression of peroxisomal beta-oxidation enzymes is intriguing. Regrettably, only an assumption can be made on the underlying mechanism and thus this line of thinking is unfinished. In the second part, the positive effect of TMEM135 deletion on ob/ob mice and fatty liver disease is shown, but the causative role of increased peroxisomal beta-oxidation and of reduced DHA levels is not demonstrated. Because TMEM135 is also expressed in other cellular compartments (mitochondria, lipid droplets), other mechanisms can also play a role. The lack of conclusions that can be drawn from both parts of the manuscript is very evident in the abstract.

Response: We appreciate the kind words from Reviewer 1 on our efforts to revise and improve our manuscript based on their comments. From their feedback on our revised manuscript, we recognize that we did not fully connect our findings in Figures 1-4 to the data presented in Figures 6-9. To produce a more cohesive and conclusive manuscript, we have added new data and made substantial textual edits. The revisions we have made to our resubmitted manuscript are summarized for Reviewer 1 in the below bullet points:

- 1) We have added new data to strengthen our conclusions in our resubmitted manuscript. We have added the quantification of peroxisomes using both IHC and ICC methods in *Tmem135* mutant tissues and cells (see **Figures 4-6**). We have also included new data showing the important role of PPAR α signaling on the peroxisomal changes observed in the *Tmem135* mutant mice (see **Figure 6**). This new data has helped to strengthen our conclusions that we made in the previous edition of our manuscript.
- 2) We have revised the text of our abstract, introduction, results, and discussion section to better connect our findings between Figures 1-4 (**now Figures 1-6**) to Figures 6-9 (**now Figures 7-9**). We have also developed a hypothesis that is explained in our abstract and discussion as well as illustrated in our **Figure 10** to explain and connect our findings. We have also included more evidence from the literature in order to support our conclusions as requested by Reviewer 1. Lastly, we have also included a section to the discussion of our resubmitted manuscript on the limitations of our study. We believe these edits have made the text in our resubmission more cohesive and conclusive for readers.

Abstract

Mutation in TMEM135 causes both a reduction in DHA and an increase in peroxisomal beta-oxidation enzymes: without any further explanation or hypothesis, this is contradictory according to the present knowledge and very confusing. In addition, it remains unclear whether the positive effect of the TMEM135 mutation in ob/ob mice is also mediated by a reduction in DHA

Response: We have made extensive revisions to the abstract and other parts of our manuscript to reflect one cohesive study. We have added a hypothesis to explain why the *Tmem135* mutant mice have reduced DHA and increased peroxisomal beta-oxidation enzyme concentrations. We have also clarified the positive effect of the *Tmem135* mutation on leptin mutant mouse phenotypes. We do not believe the positive effect originates from the reduction of DHA due to the *Tmem135* mutation but rather the positive effect stems from the increase of peroxisomes and their beta-oxidative functions due to the activation of PPAR α signaling. Our data is consistent with a previous study that showed a PPAR α agonist treatment ameliorated the disease phenotypes of leptin mutant (*Lep^{ob/ob}*) mice (PMID: 15917863). Our revised abstract now reads:

Transmembrane protein 135 (TMEM135) is thought to participate in the cellular response to increased intracellular lipids yet no defined molecular function for TMEM135 in lipid metabolism has been identified. In this study, we performed a lipid analysis of tissues from Tmem135 mutant mice and found striking reductions of docosahexaenoic acid (DHA) across all Tmem135 mutant tissues, indicating a role of TMEM135 in the production of DHA. Since all enzymes required for DHA synthesis remain intact in Tmem135 mutant mice, we hypothesized that TMEM135 is involved in the export of DHA from peroxisomes. The Tmem135 mutation likely leads to the retention of DHA in peroxisomes, causing DHA to be degraded within peroxisomes by their beta-oxidation machinery. This may lead to generation or alteration of ligands required for the activation of peroxisome proliferator-activated receptor α (PPAR α) signaling, which in turn could result in increased peroxisomal number and beta-oxidation enzymes observed in Tmem135 mutant mice. We confirmed this effect of PPAR α signaling by detecting decreased peroxisomes and their proteins upon genetic ablation of Ppara in Tmem135 mutant mice. Using Tmem135 mutant mice, we also validated the protective effect of increased peroxisomes and peroxisomal beta-oxidation on the metabolic disease phenotypes of leptin mutant mice which has been observed in previous studies. Thus, we conclude that TMEM135 has a role in lipid homeostasis through its function in peroxisomes.

Introduction

Line 90-92 and sentence of subsequent paragraph should be reorganized

Response: We have edited this paragraph and now it reads:

*Transmembrane protein 135 (TMEM135) is a 52 kilodalton protein with five transmembrane domains that is important for murine retinal and cardiac health¹³⁻¹⁵. Multiple proteomic studies have identified TMEM135 as a key component of peroxisomes¹⁶⁻¹⁹, but it is also present on other organelles including mitochondria^{13,20} and lipid droplets^{20,21}. While no study has defined the molecular function of TMEM135 on these organelles, it has been speculated that TMEM135 may play a role in the cellular stress response to increased intracellular lipids²⁰. This cellular stress response may impinge on mitochondrial dynamics^{13,15,22-25}, energy expenditure²⁰, and cholesterol degradation²⁶ as these pathways are affected in cells with altered TMEM135 function. In support of the hypothesis that TMEM135 participates in maintaining lipid homeostasis, we have recently found a mutation in the murine *Tmem135* gene increased the expression of genes involved in lipid metabolism in the murine retina²⁷, an organ with unique lipid demands²⁸.*

Results

Line 128: when analyzing acyl side chains of the lipids, only strong decreases in DHA are stated. Although it is OK to focus on this PUFA in the rest of the text, at least it should be mentioned what happens with the levels of other PUFA (including AA) and saturated fatty acids.

Response: We have included additional sentences in the results section discussing our acyl side chain analysis. We also emphasized how the changes in lipids containing other fatty acids such as arachidonic acid and saturated fatty acids occurs in a tissue-specific manner. The sentences we added are as follows:

*We found a large proportion of lipids containing DHA (C22:6n3) decreased across all the *Tmem135*^{FUN025/FUN025} tissues used in this study (Fig. 1b). We also observed modifications of lipids containing other fatty acids including C16:0, C16:1, C18:0, C18:1, C18:2, C20:3, C20:4, and C22:5 in the *Tmem135*^{FUN025/FUN025} tissues (Table S2). However, the changes of these lipids appeared to occur in a tissue-specific manner (Table S2).*

Figure 3d: the peroxisomal enzymes ACOX1, DBP and ACAA1 are processed into smaller fragments after their import in peroxisomes. In the figure of the western blot, it is not mentioned which is the size of the protein that is detected.

Response: We agree with Reviewer 1 that ACOX1, DBP, and ACAA1 are processed into smaller fragments after their import into peroxisomes as observed on our Western blot images (see **uploaded Microsoft Excel Raw Data file**). For this study, we have analyzed the immunobands that correspond to the full-length forms of these proteins. However, we do observe higher intensities of the smaller immunobands for ACOX1, DBP and ACAA1 in the *Tmem135* mutant mice, indicating more ACOX1, DBP and ACAA1 protein within peroxisomes in the *Tmem135* mutant mice (see **uploaded Microsoft Excel Raw Data file**). To clarify the size of the immunoband analyzed in this study, we have included the protein size of the analyzed immunoband next to its representative image in every figure that includes Western blot data (see **Figures 3, 4, 6, 7, 9, S2, S3, S4, S5, S6, S8, S10, S11, and S12**).

Figure 4: from the images and the western blot, it is impossible to deduce that there are more peroxisomes present per cell and thus to conclude that there is peroxisome proliferation. Much larger magnifications are needed in order to visualize individual peroxisomes such that they can be counted. ICC is usually better suited to check peroxisomal numbers than IHC.

Response: We agree with Reviewer 1 that our conclusions on the role of TMEM135 in peroxisome proliferation would be better supported by the quantification of peroxisomes by either IHC or ICC. We counted the number of peroxisomes per cell through IHC as suggested by Reviewer 1. We collected 60X magnification images of WT, *Tmem135* TG, and *Tmem135* mutant fibroblasts that were labeled for PEX14 (**Figure 5a**). We used the ‘Analyze Particles’ function in ImageJ to quantify the number of peroxisomes in these cells as this method was previously used to quantify the peroxisome number in mouse embryonic fibroblasts (PMID: 33244184). We found increased peroxisomes in the *Tmem135* mutant fibroblasts and decreased peroxisomes in the *Tmem135* TG fibroblasts (see **Figure 5b**).

We also collected larger magnified images of our liver sections (i.e. 100X instead of 60X) that were also labeled for PEX14 (**Figure 4a**). Using the same analysis pipeline as described above, we validated our in vitro results by observing increased peroxisomes in the *Tmem135* mutant livers and decreased peroxisomes in the *Tmem135* TG livers (see **Figure 4b**).

Lastly, we have included additional data showing the important role of PPAR α signaling on the peroxisomal proliferation observed in the *Tmem135* mutant mice. Using a mouse genetics approach, we crossed *Tmem135* mutant mice with *Ppara* knockout mice to evaluate the levels of peroxisomes. We also found the genetic ablation of *Ppara* prevents the increase of peroxisome number in the *Tmem135* mutant livers (see **Figure 6b**). We have also included the quantification of PEX14-labeled peroxisomes in WT and *Ppara* knockout liver sections to our resubmitted manuscript as this strengthens our conclusions that activation of PPAR α signaling contributes to the increases of peroxisome number in the *Tmem135* mutant mice (see **Figure 6b**).

In summary, we have added the quantification of peroxisome number using both IHC and ICC to our resubmitted manuscript as recommended by Reviewer 1. We have also added more data on the role of PPARa on the peroxisomal proliferation observed in the *Tmem135* mutant mice. We conclude based on this data that the *Tmem135* mutation causes peroxisome proliferation in mice that is partly mediated through PPARa signaling.

Figure 5: CPT1A is reduced in TMEM135FUN025/FUN025 livers. It is well known that CPT1A is a PPARa target gene (PMID: 20638986, PMID: 29795111 and ref 59 of the manuscript). How is this reconciled with the claim that PPARa is activated? What was the expression of CPT1A in the TMEM135/PPARa knockouts? Because of the opposing changes in CPT1a and CACT and other mitochondrial import proteins that are unaffected, this paragraph is not very informative. To further prove PPARa activation in liver the ER enzyme CYP4A10, that is strongly regulated by PPARa, would be a good option.

Response: We appreciate the studies provided by Reviewer 1. Our genetic study combining the *Tmem135* mutation and *Ppara* deficiency has proved that PPARa activation occurring in *Tmem135* mutant mice is at least partly responsible for phenotypes observed in these mice. As requested by the reviewer, we also evaluated the protein level of CYP4A10 and found that it was significantly increased in *Tmem135* mutant livers compared to WT livers. Additionally, data from *Tmem135^{FUN025/Fun025}/Ppara^{-/-}* mice indicated that the increase of CYP4A10 in *Tmem135* mutant livers resulted from the activation of PPARa (see **Figure S6**). We have included this data in our resubmitted manuscript as it strengthens our conclusion that PPARa signaling is activated in *Tmem135* mutant mice. However, each PPARa target gene may be differentially regulated not only by PPARa but also through other mechanisms. The decreased CPT1A protein levels in the *Tmem135* mutant liver compared to those in WT livers may be occurring either from a PPARa-independent regulatory mechanism or from an impaired function of TMEM135 on mitochondria. First, there are mechanisms involving epigenetic and posttranslational modifications that can regulate the expression of the CPT1A protein (PMID: 31900483). Second, TMEM135 is known to colocalize with mitochondrial membranes, and the *Tmem135* mutation has been shown to have profound effects on mitochondrial homeostasis (PMID: 27863209, 30102730, 32515224, 33064130, and 34201955). We have added these sentences to our ‘Limitations of the Study’ section in the discussion of our resubmitted manuscript. Nonetheless, we did examine the expression of CPT1A in *Tmem135^{FUN025/Fun025}/Ppara^{-/-}* livers as prompted by the reviewer and found that it was decreased compared to *Tmem135* mutant livers (see **below data**). This data indicated that CPT1A expression in the *Tmem135* mutant liver is dependent on PPARa.

Western blot analysis of carnitine palmitoyltransferase 1A (CPT1A) using livers from 3-month-old *Tmem135^{FUN025/FUN025}* (FUN025) and *Tmem135^{FUN025/FUN025}/Ppara^{-/-}* (FUN025/Ppara^{-/-}). Data is presented mean ± SD. Dots represent individual data points. ** indicates post hoc Tukey test for a P<0.05 significance following a significant difference detected by Student's two-way T-test.

Since Reviewer 1 did not believe the inclusion of the paragraph describing Figure 5 in the previous version of our manuscript was very informative, we have removed this paragraph and moved this data to our **Figure S10**. We felt it was important to still include this data in our resubmitted manuscript since this data was specifically requested by Reviewer 2 during the last round of peer review.

Line 235: cholesterol is reduced in double TMEM/ob/ob mice. How are these data explained in light of changes in peroxisomes?

Response: We believe that the decrease of plasma cholesterol in the double *Tmem135* and leptin mutant mice originates from a decrease of secreted very-low density lipoproteins (VLDLs) in these mice. This notion is supported by a decrease of plasma APOB100 in the double *Tmem135* and leptin mutant mice since APOB100 is the main protein constituent of VLDL (**see Panel g of Figure 7**). The decreased plasma VLDL may occur as a result from the activation of PPAR α due to the *Tmem135* mutation (**see Figure 6**). For example, the activation of PPAR α signaling can upregulate the expression of the VLDL receptor and decrease the amount of plasma VLDL in mice (PMID: 24899625).

Line 288: what are ‘numbers’ referring to?

Response: We have changed numbers to proteins. This sentence now reads:

*This data verified that the *Tmem135* mutation reduces DHA-containing lipids but increases peroxisomal proteins in *Lep^{ob/ob}* mice.*

Discussion:

Line 303 -306. Several mouse models are mentioned in which DHA is lacking. If peroxisomes are essential for DHA synthesis, are there no mouse models in which peroxisomal beta-oxidation is deficient?

Response: There are multiple mouse models that either lack peroxisomes or enzymes required for peroxisomal beta-oxidation, some of which display decreases in DHA. However, we are unaware of the retinal phenotypes of these animals and have not included them in this paragraph. The purpose of this paragraph is to highlight mouse models with DHA deficiencies that show global lipid profile changes and retinal degeneration. Instead of including mouse models with peroxisomal beta-oxidation abnormalities to this section, we have included information on mouse models with peroxisomal abnormalities where we discuss the possible origins of the DHA deficiencies in *Tmem135* mutant mice (**see below for these sentences**).

Line 319-321: can the authors be more specific how the phenotypes of mouse models with beta-oxidation deficiency differ from TMEM135 deficient mice?

Response: We apologize that our statement was vague. Our intention behind this statement was to describe differences between the *Tmem135* mutant mouse and previous mouse models that have peroxisomal abnormalities. As already noted, the *Tmem135* mutant mouse has increased peroxisomes and augmented peroxisomal beta-oxidation enzymes along with their decreased DHA concentrations. This mouse model differs from other previously published mouse models of peroxisome abnormalities, which have DHA reductions caused by either absent peroxisomes or peroxisomal beta-oxidation enzymes. We have included a section to our discussion highlighting these mouse models. This section of our resubmitted manuscript reads:

*There lies the key difference between *Tmem135* mutant mice and other mouse models displaying DHA deficiencies from peroxisomal abnormalities. Mice with peroxisomal biogenesis defects*

that are unable to produce functional peroxisomes such as peroxisome biogenesis factor 2⁶⁷ and peroxisome biogenesis factor 5 knockout mice⁶⁸ have reduced DHA concentrations. Also, mice with peroxisomal beta-oxidation defects such as Acox1⁶⁹ and multifunctional protein 2 (also known as DBP) knockout mice⁷⁰ have decreased levels of DHA. However, these mice obviously lack peroxisome functions in general or peroxisomal beta-oxidation capacity, while Tmem135 mutant mice retain them. These unique characteristics may underlie the increase in peroxisomes that occurs in Tmem135 mutant mice as described below.

Line 383-385: the suggestion that mitochondrial beta-oxidation would be down regulated when peroxisomal beta-oxidation is upregulated, is not very likely. These two beta-oxidation pathways are not redundant and serve different functions. For the breakdown of the common fatty acids (C16 -C18), mitochondrial beta-oxidation is much more adapted and, in contrast to peroxisomal beta-oxidation, energy generating. As mentioned before, normally peroxisomal and mitochondrial beta-oxidation are coordinately upregulated by PPARalpha (see also refer 59).

Response: We agree with Reviewer 1 and have removed this suggestion from our manuscript.

Minor comments

Line 150: ‘determined’

Response: We have corrected this sentence and it now reads:

*To decipher the role of TMEM135 in cellular DHA metabolism, we harvested livers from 2.5-month-old WT and Tmem135^{FUN025/FUN025} mice and **determined** the expression level of key components of the Sprecher pathway of DHA synthesis (Fig. 3a)³².*

Line 321: based ‘off’ these results

Response: This sentence has been removed from our resubmitted manuscript.

There are more textual flaws

Response: We apologize to Reviewer 1 for textual flaws in the previous version of our manuscript. In our resubmission, we have carefully proofread this version and tried correcting all textual mistakes. We are willing to make further corrections if there are any textual errors that we have missed in our resubmitted manuscript.

Response to Reviewer #2’s comments:

This is a resubmission Pinel et al. On the first submission, we had several concerns and questions that have been reasonably addressed. One major concern remaining. Some of the data, as presented will elicit a variety of controversies in the field.

Response: We have made extensive edits to the text of our resubmitted manuscript to lessen the controversies that may be associated with it in the field.

1. Omega-3 long chain PUFA supplementation has a beneficial effect on obesity, diabetes, cardiovascular disease, thermogenic function of adipocytes, and counteract the effects of omega-6 long chain PUFA. So, in the clinical realm, we try to supplement with Omega-3 to mitigate the ill

effects of obesity and excess fat. The major concern here is, why in the setting of a this TMEM135 mutation (presumed knockdown), reduced DHA is beneficial?

Response: We agree that DHA supplementation has a beneficial effect on a number of pathological conditions, ranging from obesity to cardiovascular disease. For example, DHA is incredibly important for nervous tissues, and the decreased DHA in the *Tmem135* mutant mice most likely explains why these mice develop retinal degeneration with severe neuroinflammation.

However, the role of DHA in other tissues is not fully understood. For instance, although the production of DHA in the liver and its consequent supply to the plasma must be critical, DHA may not be necessary for the function of the liver itself. Our results may seem controversial considering the generally accepted notion that DHA is necessary and beneficial for every tissue. However, our results may point to a tissue-dependent reliance for DHA that is required for normal functioning. We believe that our results may open a door to re-evaluate the role of DHA in a tissue-specific manner in future studies.

To repeat, we do not believe that the protection associated with the *Tmem135* mutation in the leptin mutant mice is from the reduction of DHA nor that reduced DHA is beneficial. Rather, we suspect that the protection comes from activation of the PPAR α signaling pathway and subsequent increase in peroxisome number and beta-oxidation in *Tmem135* mice. In support of this hypothesis, we observed that increases of peroxisomes and their beta-oxidation enzymes in the *Tmem135* mutant liver are at least partly mediated by the activation of PPAR α signaling (see **Figure 6**), and that the *Tmem135* mutation increases peroxisomes and their proteins in leptin mutant mice (**Figure 9**).

2. I again agree with publication of the data, although the DHA story is still counterintuitive. In any event, the data seem true, novel, and relevant to the clinical interest, in terms of a role for TMEM135 in lipid accumulation in the setting of a variety of illnesses.

Response: Future studies will need to be performed to assess the function of TMEM135 in a variety of pathological settings. This may be important in determining the therapeutic effectiveness of targeting TMEM135 in various disease situations.

3. As the authors know the role of DHA's in health and illnesses has emerged as a critical clinical piece. Nevertheless, the mechanisms of how Omega-3 long chain PUFAs help with reduced inflammation or help with obesity and heart disease are still unclear. So, altered DHA metabolism may be totally incidental, and not the true cause for TMEM135 knockdown help.

Response: We agree with Reviewer 2 on their statement and removed our statements on the role of altered DHA metabolism being responsible for the protection of the *Tmem135* mutation on the leptin mutant mouse phenotypes. We believe the protection could be attributed to activation of PPAR α signaling causing increased peroxisome number and beta-oxidation.

4. In essence, this is reasonable publishable piece of research, but the authors have not completely delineated a clear mechanism. There are several unanswered questions. The authors cannot address them all in one manuscript, but few basic questions should be answered. For instance, do the authors know if caloric expenditure and food intake are altered in these mice? Is this part of future studies?

Response: We have carried out a study to examine food intake and energy expenditure in wild-type and *Tmem135* mutant mice and found no change in either of these parameters between those genotypes. Interestingly, we observed a decreased respiratory exchange ratio in *Tmem135* mutant mice relative to wild-type that indicates increased fatty acid oxidation in the *Tmem135* mutant mouse. We have included

this data in our **Figure S11**. Further studies need to be completed to fully interrogate the effect of the *Tmem135* mutation in leptin mutant mice on feeding behavior, substrate utilization, and energy expenditure. However, these are outside the scope of this manuscript and will be examined in the future.

5. Perhaps in the discussion section, the authors can better highlight the potential controversies and soften the tone, indicating that mitigating effects for the ob/ob_TMEM135 knockdown is perhaps NOT through alteration of DHA metabolism. Those non-DHA ideas will be further explored in subsequent manuscripts.

Response: We agree with Reviewer 2 that the protective nature of the *Tmem135* mutation may not be directly through the alteration of DHA metabolism. Instead, the mitigating effects may be through the activation of PPAR α and resulting increase in peroxisome number due to the *Tmem135* mutation based on new data that we provided in **Figure 6** of our resubmitted manuscript. It is also possible that the protection is from the effect of the *Tmem135* mutation on other organelles, which we discussed in our 'Limitations of the Study' section of our discussion. We are undertaking studies to determine the origin of the protection from the *Tmem135* mutation on leptin mutant mouse phenotypes.

6. In summary, the paper can be accepted with some revision of the discussion section.

Response: We greatly appreciate the time and efforts provided by Reviewer 2 in assessing our manuscript.

REVIEWERS' COMMENTS:

Reviewer #1 (Remarks to the Author):

The authors substantially improved the manuscript, which now reads as a coherent piece of work. In particular, the abstract reflects a clear hypothesis and rationale to combine all the presented results. I do not have further comments.